# Dietary Choline Alleviates High-Fat Diet-Induced Hepatic Lipid Dysregulation via UPRmt Modulated by SIRT3-Mediated mtHSP70 Deacetylation

**DOI:** 10.3390/ijms23084204

**Published:** 2022-04-11

**Authors:** Yu-Feng Song, Hua Zheng, Zhi Luo, Christer Hogstrand, Zhen-Yu Bai, Xiao-Lei Wei

**Affiliations:** 1Key Laboratory of Freshwater Animal Breeding, Ministry of Agriculture, Fishery College, Huazhong Agricultural University, Wuhan 430070, China; zhenghua1@webmail.hzau.edu.cn (H.Z.); luozhi99@mail.hzau.edu.cn (Z.L.); baizhenyu12345@163.com (Z.-Y.B.); xiaolei1205@webmail.hzau.edu.cn (X.-L.W.); 2Laboratory for Marine Fisheries Science and Food Production Processes, Qingdao National Laboratory for Marine Science and Technology, Qingdao 266237, China; 3Department of Nutritional Sciences, School of Medicine, King’s College London, Franklin-Wilkins Building, 150 Stamford Street, London SE1 9NH, UK; christer.hogstrand@kcl.ac.uk

**Keywords:** SIRT3, UPRmt, hepatic lipid dysregulation, choline, high-fat diet

## Abstract

The mitochondrial unfolded protein response (UPRmt) is known as a conservative mechanism in response to mitochondrial dysfunction. Thus, based on UPRmt, this study was conducted to determine the mechanism of a high-fat diet (HFD) inducing mitochondrial dysfunction and its role in stimulating hepatic lipid dysregulation. The choline-activated alleviating effect was also evaluated. In vivo, yellow catfish were fed three diets (control, HFD, and HFD + choline diet) for 10 weeks. In vitro, hepatocytes isolated from yellow catfish and the HepG2 cell line were cultured and incubated with fatty acid (FA) for 48 h. (1) HFD-induced mitochondrial dysfunction via SIRT3/mtHSP70-mediated UPRmt. HFD inhibited the subcellular localization of SIRT3 into the mitochondrion, resulting in the up-regulating of mtHSP70 acetylation via lysine residues 493 and 507. The mtHSP70 acetylation promoted the stability of mtHSP70, which then led to the UPRmt and further mitochondrial dysfunction. (2) SIRT3/mtHSP70-mediated UPRmt regulated HFD/FA-induced hepatic lipid dysregulation. SIRT3/mtHSP70-mediated UPRmt reduced FA ß-oxidation via mitochondrial dysfunction and then led to lipid dysregulation. Additionally, the mtHSP70–ACOX1 interaction was confirmed. (3) Choline alleviated HFD-induced UPRmt via up-regulating the localization of SIRT3 into the mitochondrion, which in turn led to the subsequent ameliorating effect on HFD-induced hepatic lipid dysregulation. Through SIRT3-mediated mtHSP70 deacetylation, dietary choline alleviates HFD-induced hepatic lipid dysregulation via UPRmt. This provides the first proof of acetylation regulating UPRmt and the crosstalk between UPRmt and FA ß-oxidation.

## 1. Introduction

Currently, as the most common chronic liver disease for human beings [1], high-fat diet (HFD)-induced hepatic steatosis is increasing rapidly worldwide [2]. Steatosis is characterized by hepatic lipid dysregulation. Hepatic lipid accumulation is regulated by a complex network involving lipogenesis, dietary fatty acid (FA) uptake, FA oxidation, and lipid transport/secretion [3]. Since mitochondria (MT) are the main site of FA ß-oxidation, mitochondrial dysfunction is also recognized as a common sign for HFD-induced hepatic steatosis. This phenomenon has been found in our previous study on fish [4]. However, the mechanism of HFD causing mitochondrial dysfunction and its role in hepatic lipid dysregulation still remains to be explored. The mitochondrial unfolded protein response (UPRmt) is a conserved adaptive mechanism in response to mitochondrial damage [5]. Importantly, the connection between UPRmt and steatosis has been revealed [6]; however, the molecular mechanism remains poorly understood. Therefore, UPRmt should be a good viewpoint from which to understand HFD-induced mitochondrial dysfunction and also a potential target for intervening and preventing hepatic steatosis.

As an adaptive stress response pathway, the UPRmt acts similarly to the UPR of endoplasmic reticulum (ER) [7]. While the UPR was the first to be identified and has been well known, the activation of the UPRmt has only recently begun to be appreciated [8]. Various types of mitochondrial stress could lead to its misfolding and aggregation in the MT. Cells sense and respond to mitochondrial damage by activating UPRmt, which is dedicated to maintaining mitochondrial homeostasis. UPRmt includes the induction of mitochondrial chaperones (HSP10, HSP60, and mtHSP70) and proteases (LONP1 and CLPP), which assist in proper protein folding and promote the clearance of misfolded proteins [9]. Although significant progress in investigating UPRmt has been made [10], the mechanism of HFD inducing UPRmt remains to be investigated. Moreover, the molecular signal transduction network that UPRmt triggers hepatic lipid dysregulation has not been established.

By now, UPRmt is considered to be a transcriptional response aiming to promote organelle-specific protein homeostasis through mitochondrial chaperones and proteases. However, recently, increasing evidence has shown that post-translational modifications also play an important role in maintaining the function and stability of mitochondrial chaperones and proteases. For the HSP70 family of proteins, a vast array of post-translational modifications (PTMs) on HSP70 family proteins has been uncovered, including acetylation, ubiquitination and AMPylation. Importantly, this complex pattern of chaperone PTMs is now known as the “chaperone code” [11], implying the potential role of post-translational modifications in mediating mtHSP70. Currently, the primary mitochondrial deacetylase SIRT3 is known as a major coordinator for orchestrating UPRmt [12] and also mediates mitochondrial protein acetylation [13]. However, the role of SIRT3-regulated acetylation in mediating UPRmt is unknown. Thus, related studies will provide the first proof that UPRmt is also controlled by post-translational modifications, not just a transcriptional mitochondrial-to-nuclear cross talk.

As a lipid-lowering agent [14], choline is closely associated with mitochondrial function [15]. A study demonstrated that a choline-deficient diet could induce abnormal mitochondrial functions and then cause fatty liver disease [16]; however, the molecular mechanism remains largely unknown. Furthermore, choline could attenuate cardiac dysfunction by UPRmt [17], suggesting the implication of UPRmt in choline-mediating cellular processes. However, the role of UPRmt in choline-ameliorating mitochondrial dysfunction and hepatic lipid dysregulation remains to be investigated.

Fish are the by far the largest group of vertebrates in the world. During the evolution of vertebrates, the entire genome was duplicated twice. However, fish have a third genome duplication, called the fish-specific genome duplication (FSGD) [18]. FSGD also has been observed in yellow catfish (*Pelteobagrus fulvidraco*), a freshwater teleost widely distributed in China and other countries [19]. Some duplicated genes evolved new functions that in turn resulted in novel regulatory mechanisms [16]. Moreover, yellow catfish frequently exhibit excessive hepatic lipid accumulation after HFD feeding [20]. Therefore, using yellow catfish as a model, we hoped to find some novel regulatory mechanism of lipid metabolism. Our previous studies found that HFD caused excessive hepatic lipid accumulation and coincided with massive swelling of the mitochondrion, suggesting the close link between mitochondrial dysfunction and hepatic lipid dysregulation. However, the underlying mechanism remains unknown. Additionally, choline as a widely used lipid-lowering agent has shown potential for a role in alleviating mitochondrial dysfunction. Thus, we are interested in the mechanism of HFD-induced mitochondrial dysfunction stimulating hepatic lipid dysregulation and the choline-activated alleviating effects. Our present study revealed a novel regulatory mechanism of HFD inducing hepatic lipid dysregulation via UPRmt and the role of SIRT3-mediated mtHSP70 deacetylation on the induction mechanism of UPRmt. This provided the first evidence of post-translational modification mediating UPRmt and the direct regulatory link between UPRmt and FA ß-oxidation, which may help inform the development of novel targets for the treatment of hepatic steatosis.

## 2. Materials and Methods

### 2.1. Experimental Treatments

The present experiments were performed according to the institutional ethical guidelines of the Huazhong Agricultural University (HZAU) for the care and use of laboratory animals. The work has received approval for research ethics from HZAU and a proof/certificate of approval is available upon request (Ethical Approval No. HBAC20091138).

#### 2.1.1. In Vivo Experiment

Ingredients and feed formulation for three experiments diets are shown in Appendix A. Fish oil and corn oil were added at a ratio of 1:1. Dietary fat levels were 10.3% for the control group, 14.5% for the high-fat diet (HFD) group, and 14.3% for high fat group + choline (HFD + CH) group, respectively. Choline chloride was added to the test diets at the expense of cellulose to provide concentrations of 563.4 (Control), 578.9 (HFD), and 1652.3 (HFD + CH) mg of choline per kg diet. Supplemental levels were confirmed by analysis. The lipid and choline supplemental levels were determined according to our previous studies [20,21]. Each diet was fed to three replicate tanks of yellow catfish (mean initial weight: 3.83 ± 0.01 g, mean ± SEM), with 25 fish per tank. Feeding and culture management were similar to those in our recent studies [4].

At the end of the 10-week experiment, yellow catfish were counted and weighed in bulk. Their survival, specific growth rate (SGR), and weight gain (WG) were determined. The liver tissues from three fish of each tank were sampled for histological, histochemical and ultrastructural observation, respectively. The liver samples from other fish were frozen immediately in liquid nitrogen for other analysis, including contents of TG and ATP, enzymatic activities, and gene and protein expression.

#### 2.1.2. In Vitro Experiment

Primary hepatocyte culture and treatment: Yellow catfish (mean weight: 6.83 ± 0.24 g, mean ± SEM) were obtained from the in vivo experiment after two weeks of acclimation. Primary hepatocytes were isolated and cultured as previously described [4]. In order to explore the mechanism of FA and choline-influencing UPRmt and lipid metabolism, we used FA and choline to incubate the primary hepatocytes of yellow catfish.

The total concentration for oleic acid (OA) and palmitic acid (PA) as a ratio of 1:1 was 0.5 mM for the FA group and FA + CH group, according to our recent publications [4,20]. They were dissolved in DMSO before addition. The total concentration of choline chloride was 0.5 mg/L for the FA + CH group based on the levels of choline chloride in M199 and the cell viability in our pilot experiment. The three experimental treatments were designed: the control (without extra addition), FA (0.5 mM), and FA and choline combination (0.5 mM of FA + 0.5 mg/L of choline). Additionally, there were specific inhibitors for protein degradation [(R)-MG-132] or synthesis (CHX). The cells were incubated at 28 °C for 48 h. For each cell culture, a pool of cells from three fish was used.

HepG2 cell line culture and treatment: HepG2 cell lines were cultured as previously described [22]. Using the HepG2 cell line, mtHSP70 acetylation levels, its related acetylation site, SIRT3–mtHSP70 interaction, and mtHSP70 with ACOX1 were identified by transfecting mtHSP70, SIRT3, and ACOX1 expression vectors.

### 2.2. Sample Analysis

#### 2.2.1. H&E, Oil Red O, Bodipy 493/503 Staining, and TEM Analysis

Hematoxylin and eosin (H&E) and Oil Red O staining tests and a TEM analysis were conducted according to the methods described in our previous study [4]. Bodipy 493/503 staining was performed to test intracellular lipid droplets as described previously [4].

#### 2.2.2. Determination of mRNA Levels through Real-Time Q-PCR

Analyses on gene transcript levels were conducted through the real-time quantitative PCR (RT-qPCR) method described before [4]. The primer sequences used in this analysis are given in Appendix A. A set of eight housekeeping genes (β-actin, GAPDH, EF1A, 18S rRNA, HPRT, B2M, Tuba, and RPL17) were selected from our transcriptome database in order to test their transcription stability. Our pilot experiment indicated that β-actin and GAPDH (M = 0.27) showed the most stable levels of expression across the experimental conditions, as suggested by geNorm [23]. Thus, the relative expression levels were normalized to the geometric mean of the combination of β-actin and GAPDH and calculated using the 2^−ΔΔCt^ method.

#### 2.2.3. Contents of ATP, Triglyceride (TG), and Acetyl-CoA

ATP was measured using an ATP Colorimetric/Fluorometric Assay Kit (Beyotime, Haimen, China) as described in our previous study [23]. Briefly, cells were lysed and centrifuged at 12,000 RPM for 5 min. The supernatant (100 µL) was transferred to a 24-well plate and then mixed with ATP detection working solution (100 µL). Luminescence signals were measured by a microplate reader (BMG Labtech, Offenburg, Germany). The protein concentration of each group was also determined using a Coomassie protein assay reagent (Thermo Fisher Scientific, Eugene, OR, USA). The relative ATP level was expressed as ATP value/protein value. The contents of TG were determined according to our previous studies [24] using a TAG Assay Kit (Jiancheng Biotech, Nanjing, China). The acetyl-CoA concentrations were determined by using HPLC as described previously [25].

#### 2.2.4. SIRT3 Deacetylase Activity, Activities of ACOX1 and HSP70-ATPase, and Mitochondrial Palmitic Acid (PA) β-Oxidation Efficiency

After the isolation and purification of the mitochondria from the hepatocytes of yellow catfish the mitochondrial SIRT3 deacetylase activity was assayed using a deacetylase colorimetric activity assay kit according to the manufacturer’s instructions (Biomol International, Plymouth Meeting, PA, USA) as described in Xue et al. [26]. ACOX1 enzyme activity was determined as described previously [27]. Briefly, reactions were initiated by adding 5 μL of ACOX1 protein and analyzed at 30 °C. Activities were measured spectrophotometrically by recording the oxidation of 4-aminoantipyrine (4-AA)/phenol to quinoneimine dye by peroxidase (POD) at A500 nm. The activity of ACOX1 was calculated by the ratio of A500 change in a total of 15 min. Assays that approximate single turnover measurements of endogenous HSP70-ATPase activity were performed at 30 °C on pre-formed [α^32^P] ATP-HSP70 complexes according to published methods [28]. Briefly, 25 μg of HSP70 were incubated with 100 μCi of [α^32^P] ATP on ice for 30 min in complex buffer and 25 μm ATP in a final volume of 100 μL. Glycerol was added to a final concentration of 10%, and 25 μg aliquots were frozen in liquid nitrogen and stored at −80 °C for no longer than 3 weeks. To assay ATP hydrolysis, individual aliquots were rapidly thawed and added to an equal volume of complex buffer containing test compounds or Me2SO pre-equilibrated to 30 °C. The final concentration of HSP70 was ∼0.2–0.3 μm in these reactions. Mitochondrial PA β-oxidation efficiency was determined using labeled [1^−14^C] palmitic acid (PA) (PerkinElmer, Pittsburgh, PA, United States) as a substrate, as reported in a previous study [25].

#### 2.2.5. DNA Isolation and rt-PCR for mtDNA

The mtDNA was isolated according to the methods described in our previous study [4], using a commercial kit (MT DNA Isolation Kit, Biovision, Mountain View, CA, USA). The mtDNA levels were measured with an SYBR green dye-based RT-PCR assay (Thermo Fisher Scientific, Waltham, MA, USA) using an ABI PRISM 7300 sequence detection system (Applied Biosystems, Foster City, CA, USA). The primer sequence was the yellow catfish NADH dehydrogenase 1 gene (mtDNA): forward 5′-GGAGCAGTAGCCCAAACAAT-3′ and reverse 5′-AGTGATAAGGGTGCAGAGGTT-3′. Plasmid DNA with complementary DNA sequence for yellow catfish mtDNA was obtained from OriGene Technologies (SC101172; Rockville, MD, USA). Concentrations of cytoplasm mtDNA were converted to copy number via a DNA copy number calculator [29]. Plasmid DNA were diluted in tenfold serial dilutions and measured as a standard curve. All samples were measured with standards at the same time.

#### 2.2.6. Immunofluorescent Staining

The distribution of SIRT3 into the mitochondrion (MT), the acetylation level, and the co-localization of mtHSP70 and ACOX1 were analyzed through immunofluorescent staining, based on a study by Jeong et al. [30]. Generally, samples were fixed in 4% paraformaldehyde and permeabilized with 0.3% Triton X-100 (Sigma-Aldrich, St. Louis, MO, USA) at room temperature for 10 min. The samples were blocked for 30 min in PBST with 1% BSA and 22.52 mg/mL of glycine, followed by incubation with anti-SIRT3 (ab217319; abcam, Cambridge, UK), anti-acetyl Lysine (ab21623; abcam, Cambridge, UK), anti-mtHSP70 (ab2799; abcam, Cambridge, UK), or anti-ACOX1 (ab184032; abcam, Cambridge, UK) overnight at 4 °C. Our previous studies have proved that antibodies from other various species can be used on yellow catfish [2]. Then, incubation with a goat anti-rabbit IgG H&L secondary antibody (ab150079; abcam, Cambridge, UK) continued for 1 h at room temperature. The nucleus and mitochondrion were stained with DAPI and MitoTracker Red CMXRos, respectively. Fluorescent microscopy and the laser scanning confocal microscope (Leica Microsystems, Wetzlar, Germany) were used for the observation.

#### 2.2.7. Flow Cytometric Analysis

Flow cytometric analysis was performed according to our previous study [20]. In brief, a FACSort (Becton Dickinson, Sunnyvale, CA, USA) equipped with a single argon-ion laser was used according to the methods described previously [31]. Excitation was done at 488 nm, and the emission filters used were 515–545 BP (green; FITC) and 600 LP (red; PI). A minimum of 5000 cells per sample was analyzed, and data were stored in list mode. Electronic compensation was used to eliminate the bleed through of fluorescence. Data analysis was performed with the standard Lysis and CellFIT software (Becton Dickinson, Mountain View, CA, USA).

#### 2.2.8. Mitochondrial Membrane Potential (MMP) Measurement

Measurements of MMP were made by staining hepatocytes with the lipophilic substance 5,5,6,6-tetrachloro-1,1,3,3-tetraethylbenzimidazolyl carbocyanine iodide (JC-1) according to the manufacturer’s instructions, as described in our recent study [32]. This dye differentially labels mitochondria according to their membrane potential, emitting in the high orange wavelength for high MMP and in the green wavelength for low MMP. Briefly, the cells were incubated with JC-1 staining solution (5 μg/mL) for 20 min at 28 °C. Then, the cells were rinsed twice with JC-1 staining buffer and resuspended in 300 μL of JC-1 staining buffer. Two laser lines (488 nm and 552 nm) were used to collect fluorescence images (green and red fluorescence) through fluorescence laser scanning confocal microscopy (Leica, Wetzlar, Germany). The MMP was calculated as the fluorescence ratio of green to red.

#### 2.2.9. Mass Spectrometric Analysis

The LC-MS/MS analysis was used to identify the acetylation site of SIRT3, and the analysis was performed on an EAST-nLC 1200 system (Thermo Fisher Scientific, Eugene, OR, USA) coupled with an Orbitrap Q Exactive HF-X mass spectrometer (Thermo Fisher Scientific, Eugene, OR, USA) following the published procedures [33].

Raw data were processed using the integrated Andromeda search engine with FDR < 0.01 at the protein, peptide, and modification levels. Variable modifications for acetylation and oxidized methionine and fixed modifications for carbamidomethyl were included in the search.

#### 2.2.10. RNAi and Gene Transfection

To generate a SIRT3 knockdown cell, hepatocytes of yellow catfish were transfected with 103 nM of siRNA against SIRT3 from Sigma-Aldrich, St. Louis, MO, USA based on previous protocols [34]. Transfection was performed with Lipofectamine 2000 (Sigma-Aldrich, St. Louis, MO, USA). Target sequences for preparing the siRNAs of yellow catfish SIRT3 are shown in Appendix A. The transfection of siRNA was performed using the Lipo2000 transfection reagent according to the supplier’s protocol. Hepatocytes were transfected with indicated siRNAs at 30% confluence. The siRNA was added to one tube and mixed gently; 5 μL of Lipo2000 transfection reagent were added to the other tube and gently mixed. Then, the culture medium containing the siRNA was gently added to the culture solution containing the Lipo2000 transfection reagent, and the tube had a gently inverted mix. A mixture of the Lipo2000 transfection reagent and siRNA was added to each well, and the culture medium was changed after 5 h.

#### 2.2.11. Plasmid and Cell Transfection

To identify the mtHSP70 acetylation sites and SIRT3-HSP70 interaction, mtHSP70 with ACOX1, we constructed the mtHSP70, SIRT3, and ACOX1 expression vectors based on our published protocols [35]. The open reading frames of mtHSP70, SIRT3, and ACOX1 sequences of yellow catfish were subcloned into the pcDNA3.1 (+) vector with the FLAG-tag and HA-tag sequences inserted at the N-terminus of the mtHSP70, SIRT3, and ACOX1 sequences, respectively. Mutations of lysines (K) 493 or/and 507 to arginines (R) were produced in the FLAG-mtHSP70 plasmid by the Mut Express II Fast Mutagenesis Kit (Vazyme). The transient transfection of the plasmids into the HepG2 cell line was conducted using Lipofectamine 2000 (Invitrogen) based on the manufacturer’s instructions.

#### 2.2.12. Immunoprecipitation and Western Blot

Immunoprecipitation was performed to identify mtHSP70 acetylation sites, SIRT3–mtHSP70 interaction, and mtHSP70–ACOX1 interaction according to our previous study [35]. Briefly, in order to conduct the immunoprecipitation analysis, we lysed the cells in NP-40 buffer (Beyotime, Nantong, China, p0013F) with the addition of a protease inhibitor cocktail (Beyotime, Nantong, China, P1010). The cell lysate was then mixed with anti-SIRT3 (ab217319; abcam, Cambridge, UK), anti-mtHSP70 (ab2799; abcam, Cambridge, UK), anti-ACOX1 (ab184032; abcam, Cambridge, UK), anti-FLAG-tag (ab1162; abcam, Cambridge, UK), or anti-HA tag (ab18181; abcam, Cambridge, UK) at 4 °C overnight, followed by the addition of protein A/G beads (P2012; Beyotime, Nantong, China). NP-40 buffer was used to wash the immunocomplexes. Finally, the Western blot analysis was performed. Additionally, due to the hepatocytes of yellow catfish not being satisfied for the immunoprecipitation experiment, all this immunoprecipitation was conducted by using another cell line (HepG2 cell line), similar to our previous study [35].

To identify the protein levels of SIRT3, mtHSP70, ACOX1, TIMM44, GAPDH, Histone H3, Ub LONP1, and acetyl lysine, a Western blot analysis was performed according to our previous study [35]. Among these antibodies, acetyl lysine was used to test mtHSP70 acetylation after immunoprecipitation and also the immunofluorescence staining. In brief, the protein was loaded onto the SDS−PAGE gel and then transferred to the PVDF membrane. Membranes were blocked with 5% skimmed milk and then incubated with the following primary antibodies overnight at 4 °C. Membranes were then incubated with corresponding secondary antibodies: HRP-conjugated anti-rabbit IgG antibody (7074; CST), HRP-conjugated mouse anti-rabbit IgG (5127; CST), or HRP-conjugated anti-rabbit IgG antibody (light chain-specific) (93702; CST). After further washing, membranes were visualized via ECL (1705060; Bio-Rad, Hercules, CA, USA.). These antibodies from other various species, including human, rat, and mouse, were used in our previous study. Studies have shown that these antibodies are feasible for yellow catfish [22,24,35].

### 2.3. Statistical Analysis

All data were expressed as mean ± standard error of means (SEM). The normality of data distribution and the homogeneity of variances were analyzed using the Kolmogorov–Smirnov test and Bartlett’s test, respectively. Then, data were subjected to one-way ANOVA and Tukey’s multiple range tests using SPSS 19.0 software, and the minimum significance level was set at *p* < 0.05. Differences between si-NC and si-SIRT3 groups were analyzed using a Student’s *t*-test for independent samples.

## 3. Results

### 3.1. Alleviated Effects of Dietary Choline on HFD-Induced Hepatic Lipid Dysregulation

As shown in Figure 1, compared to control diet groups, HFD caused lipid dysregulation in the livers of yellow catfish. HFD increased the vacuoles in H&E, lipid droplets in Oil Red O, TG content, and the expression of genes involved in lipogenesis, but it decreased the mRNA levels of genes involved in lipolysis and FA ß-oxidation. Additionally, the downregulated protein expression and activity of ACOX1 further indicated the impaired function of FA ß-oxidation in HFD groups. Importantly, compared to the HFD group, dietary choline markedly alleviated HFD-induced hepatic lipid dysregulation.

### 3.2. The Involvement of SIRT3-Mediated mtHSP70 Acetylation in Dietary Choline Attenuating HFD-Induced UPRmt

First, we examined whether HFD could cause UPRmt in the livers of yellow catfish. As shown in Figure 2A, compared to control diet groups, TEM observation found severe mitochondrial swelling in hepatocytes in HFD groups. Meanwhile, HFD increased the mRNA abundance of UPRmt markers, including mtHSP70, HSP60, HSP10, LON P1, and CLPP, and mtDNA were also marked (Figure 2B,C). Furthermore, HFD apparently up-regulated the protein level of mtHSP70 and HSP70-ATPase activity and downregulated ATP content (Figure 2D–G). All these indicated that HFD caused severe UPRmt in the livers of yellow catfish. On the other hand, choline markedly attenuated HFD-induced UPRmt, supported by the mitochondrial morpholog, the downregulation of UPRmt marker mRNA abundance, mtDNA, mtHSP70 protein levels, and HSP70-ATPase activity.

Since the stability and activity of the chaperone can be regulated by acetylation, we next investigated whether HFD could regulate the activity of mtHSP70 via acetylation. As shown in Figure 2H,I, HFD significantly increased the mtHSP70 acetylation. We further explored the involvement of SIRT3, the primary mitochondrial deacetylase regulating HFD-mediated mtHSP70 deacetylation. In the present study, both the protein levels and activity of SIRT3 were significantly decreased by HFD (Figure 2J–L). Meanwhile, the HFD-induced alteration of mtHSP70 acetylation was also markedly attenuated by dietary choline.

### 3.3. Choline Improved FA-Decreased SIRT3–mtHSP70 Interaction via Mediating the Localization of SIRT3 into MT

To further elucidate the mechanism of SIRT3 mediating mtHSP70, the SIRT3–mtHSP70 interaction was performed. First, in order to test the subcellular localization of SIRT3, the extraction and purification of mitochondrion, cytoplasm, and nucleus were conducted by centrifugation and density gradient method. Using a marker protein for mitochondrion, cytoplasm, and nucleus, separately, the subcellular localization analysis indicated the SIRT3 mainly located in the MT in the hepatocytes of yellow catfish (Figure 3A). Then, immunofluorescent observation showed that choline improved subcellular localization of SIRT3 into MT, but FA showed contrary effects (Figure 3B). Further investigation found that SIRT3 coprecipitated with mtHSP70 in HepG2 cell lines overexpressing FLAG-mtHSP70 and HA-SIRT3 (Figure 3D), suggesting the closed relation between SIRT3 and mtHSP70. Importantly, further analysis revealed that FA reduced this interaction, but choline addition attenuated this trend (Figure 3E,F), implying the involvement of FA/choline in SIRT3–mtHSP70 interaction.

### 3.4. Choline Inhibited FA Increased the Stability of mtHSP70 through SIRT3-Mediated Deacetylation

After we had determined that HFD (FA)/choline was involved in mtHSP70 acetylation and the interaction of SIRT3 with mtHSP70, we next sought to investigate the mechanism of SIRT3 mediating mtHSP70 deacetylation. Immunofluorescent staining indicated FA up-regulated whole acetylation levels in the hepatocytes of yellow catfish, but choline alleviated this trend (Figure 4A), confirming the regulatory role of FA/choline on acetylation. Considering the established interaction of SIRT3 with mtHSP70, in vitro acetylation assay furtherly revealed that SIRT3 deacetylated mtHSP70 in a concentration-dependent manner (Figure 4B), confirming that yellow catfish SIRT3 can deacetylate mtHSP70. We next determined which lysine residue of mtHSP70 was acetylated. First, the protein sequence alignment of mtHSP70 homologues showed that lysine 493 and K507 were evolutionarily conserved from fish to mammals (Appendix A), implying that K493 and K507 may be important functional acetylation sites for mtHSP70. Then, by using liquid chromatography–mass spectrometry analysis, we further confirmed that these two acetylation sites (K493 and K507) were found in the hepatocytes of yellow catfish after FA incubation (Appendix A). Lastly, the single (K493R and K507R) and double (2KR) mutants reduced the acetylation levels of mtHSP70 (Figure 4C), indicating that yellow catfish mtHsp70 could be deacetylated at the sites of K493 and K507 in response to FA.

Acetylation is a key factor for intracellular protein stability, which prompted us to test whether SIRT3-mediated deacetylation affected stability of mtHSP70. First, as shown in Figure 4D–G, in vitro under FA and choline incubation, si-SIRT3 significantly down-regulated mtHSP70 deacetylation and also promoted the protein expression level of mtHSP70. However, interestingly, the mRNA abundance of mtHSP70 was not influenced by si-SIRT3. All these implied that the stability of mtHSP70 was regulated by SIRT3-mediated deacetylation, not at the transcriptional level. Then, by using a specific inhibitor for protein degradation or synthesis, (R)-MG-132 and CHX, respectively, we further revealed that SIRT3-mediated mtHSP70 deacetylation resulted in the low protein retention of mtHSP70 via reducing protein synthesis but accelerating protein degradation (Figure 4G). These suggested an obvious regulatory mechanism for mtHSP70 acetylation in maintaining the stability of mtHSP70.

### 3.5. SIRT3-Mediated Deacetylation of mtHSP70 Is Essential for Choline-Ameliorating FA-Induced Mitochondrial Dysfunction

Our findings have indicated the underlying mechanism of SIRT3/mtHSP70 in mediating UPRmt, which prompted us to test the role of SIRT3 on choline-ameliorating FA-induced UPRmt and mitochondrial dysfunction. At first, in vitro TEM observation found choline-ameliorated mitochondrial morphology caused aggravated swelling through si-SIRT*3* (Figure 5A), implying the involvement of SIRT3 in choline improving FA-induced mitochondrial damage. Interestingly, although si-SIRT3 reversed the choline-elevated UPRmt marker at the protein expression level, including mtHSP70 and LONP1, si-SIRT3 showed no effect on the choline-reduced UPRmt marker at the transcriptional level (Figure 5B,C). This indicated that SIRT3 modulated FA/choline-influenced UPRmt mainly through deacetylation, not via transcriptional levels. Furtherly, si-SIRT3 reversed the effect of choline improving FA-reduced MMP and ATP content (Figure 5D–G), implying that SIRT3 was involved in choline ameliorating FA-induced mitochondrial dysfunction.

### 3.6. SIRT3-Modulated mtHSP70–ACOX1 Interaction Is Required for Choline-Alleviating FA-Induced Hepatic Lipid Dysregulation

Considering the close relationship between mitochondrial function and lipid metabolism, we next explored the role of SIRT3 in choline-alleviating FA-induced hepatic lipid dysregulation. First, mtHSP70 coprecipitated with ACOX1 in HepG2 cells overexpressing FLAG-mtHSP70 and HA-ACOX1 (Figure 6A). A further immunofluorescent staining assay revealed that choline reduced mtHSP70–ACOX1 interaction but FA showed contrary effects (Figure 6B), suggesting the direct regulatory association between mtHSP70 and ACOX1 and the regulatory role of FA/choline on this interaction. Then, we found that si-SIRT3 caused the loss of the choline-activated alleviating effect on FA-induced damage to mitochondrial FA ß-oxidation, including ACOX1 protein levels, mitochondrial PA ß-oxidation efficiency, and acetyl-CoA content (Figure 6C–G). Considering that ACOX1 was the rate-limiting enzyme in FA β-oxidation, the result suggested that SIRT3 regulated mitochondrial FA ß-oxidation via the interaction of mtHSP70 with ACOX1. The si-SIRT3-induced alteration of the mRNA abundance of genes involved in mitochondrial FA ß-oxidation further confirmed our above statement (Figure 6H). Smoothly and logically, Bodipy 493/503 staining indicated that si-SIRT3 also caused the loss of the choline-activated alleviating effect on FA-induced excessive lipid droplets accumulation (Figure 6I), suggesting the regulatory role of SIRT3 in hepatic lipid metabolism.

## 4. Discussion

Increasing evidence has shown that HFD is the main inducer of hepatic steatosis. Not surprisingly, our present study also found that HFD caused excessive TG accumulation and hepatic lipid dysregulation of yellow catfish, which also has been confirmed by our previous studies [4,18]. During the process of HFD causing hepatic steatosis, the two rate-limiting enzymes for lipogenesis, ACC and FAS showed sensitive reactions and significant increases in response to HFD, similar to our previous studies [20]. Meanwhile, HFD-induced hepatic lipid dysregulation was accompanied by the mitochondrial dysfunction in vivo consistent with other studies and our studies [4,36]. This present study is a continuation and extension of our previous studies. In our lab, the effect of dietary choline levels on growth performance and lipid metabolism in yellow catfish has been confirmed [21]. Thus, in this present study we designed three groups (control, HFD, and HFD + choline) to furtherly verify the choline-activated alleviating effect on the lipid metabolism. We found that dietary choline apparently relieved the HFD-induced mitochondrial dysfunction. This implied that choline alleviated hepatic lipid dysregulation via improving mitochondrial function, similar to other studies [15].

As a conservative and protective transcriptional response, UPRmt was employed to promote organelle-specific protein homeostasis in response to mitochondrial dysfunction [37]. UPRmt reduced proteotoxic stress and reestablished protein homeostasis by elevating the levels of mitochondrial chaperones and proteases [38]. As expected, we found that HFD caused the increased mitochondrial chaperones and proteases in vivo and in vitro, suggesting that HFD could induce UPRmt. At present, this rare study explores the relation between HFD and UPRmt, which make it difficult to compare our results with those of other studies. However, the fact that HFD could cause mitochondrial damage and the closed correlation between mitochondrial damage and UPRmt have been proved in previous studies [9,39]. Among these chaperones, mtHSP70 showed the most significant alteration, implying the main role of mtHSP70 on HFD inducing UPRmt. On the other hand, choline alleviated HFD/FA-induced UPRmt, suggeting that choline-activated lipid-lowering functions may work with UPRmt, similar to other studies [17].

The primary mitochondrial deacetylase SIRT3 was a significant player in regulating UPRmt [12]. SIRT3 was also involved in choline-activated ameliorating effects on HFD inducing cardiac dysfunction via UPRmt [17]. Thus, SIRT3 may have an upstream regulatory role in HFD/choline influencing UPRmt. Our results showed that HFD/FA inactivated SIRT3 both in vivo and in vitro. Consistent with our result, Kendrick et al. [40] indicated that the HFD-induced fatty liver was associated with reduced SIRT3 activity. Additionally, we further indicated that SIRT3 knockdown resulted in the loss of choline-activated ameliorating effects on UPRmt in vitro, suggesting the regulatory role of SIRT3 in UPRmt, similar to other studies [41]. Similarly, Teodoro et al. [14] suggested the association between mitochondrial dysfunction and SIRT3 inactivation in rats after high-fat meals. Then, we further showed that SIRT3 is mainly located in the mitochondrion (MT) in the hepatocytes of yellow catfish, consistent with other studies [42]. This confirmed that SIRT3 is also the primary mitochondrial deacetylase for yellow catfish. Meanwhile, FA inhibited the subcellular localization of SIRT3 into MT, but choline evidently up-regulated this process. Furthermore, we found, along with the change in the subcellular localization of SIRT3, that the interaction of mtHSP70 with SIRT3 was also inhibited by FA but improved by choline addition. Thus, we easily concluded that FA induced UPRmt by inhibiting the subcellular localization of SIRT3 into the MT, but choline alleviated this process via improving this subcellular localization.

Currently, although UPRmt is considered a transcriptional response, a mitochondrial-to-nuclear cross talk [37], growing evidence suggests that post-translational modification makes the difference for the activity and stability of UPRmt chaperones [43]. Furthermore, studies have indicated that mtHSP70 acetylation is a key regulatory modification for its chaperone activity and stability [11,44]. Given that, we have proved the SIRT3 interacted with mtHSP70. Next, we tested whether SIRT3 mediated mtHSP70 deacetylation and its role in regulating HFD/FA-induced UPRmt. First, we found that HFD/FA could increase mtHSP70 acetylation. Ma and Wood [45] reported that protein acetylation in prokaryotes could increase stress resistance. Thus, we speculated that mtHSP70 acetylation could also be the stress response for HFD/FA. Second, SIRT3 could deacetylate mtHSP70 in a concentration-dependent manner in vitro. These results led us to the conclusion that SIRT3 mediated mtHSP70 deacetylation. Then, we confirmed that FA/choline regulated SIRT3-medited mtHSP70 deacetylation. FA inhibited SIRT3-medited mtHSP70 deacetylation, but choline alleviated this process. Analogously, Kendrick et al. [40] indicated that SIRT3-mediated mitochondrial protein hyperacetylation was associated with HFD-induced fatty liver. Furthermore, the FA-mediated acetylation site of mtHSP70 was first revealed in this study. These results convinced us to conclude the implication of SIRT3 in mediating mtHSP70 deacetylation-mediated UPRmt.

Next, we tested the underlying mechanism of mtHSP70 deacetylation in regulating UPRmt. HSP70-ATPase activity is closely associated with mtHSP70 function [46]. We found the HFD-induced mtHSP70 acetylation accompanied increased HSP70-ATPase activity, further confirming the close relations between mtHSP70 acetylation and mtHSP70 function. Analogously, Aoyagi and Archer [47] suggested that reversible acetylation modulated molecular chaperone HSP90 functions. Then, we demonstrated that SIRT3-mediated mtHSP70 deacetylation resulted in low protein retention of mtHSP70 by reducing protein synthesis but accelerating degradation. Importantly, further knockdown of SIRT3 has no significant effect on the UPRmt marker at mRNA levels, including mtHSP70. Collectively, the data demonstrated that SIRT3-mediated deacetylation controlled the function and stability of mtHSP70 but not via transcriptional levels. This observation provided the first experimental evidence of post-translational modification regulating UPRmt.

We next aimed to test the role of SIRT3/mtHSP70-mediated UPRmt in mitochondrial dysfunction. Our in vitro experiment revealed that the SIRT3 knockdown resulted in aggravated mitochondrial dysfunction, similar to other studies [9]. Meanwhile, SIRT3 knockdown also aggravated mitochondrial dysfunction and led to the loss of choline-activatd ameliorating effects. These suggested the regulatory role of SIRT3 on FA/choline-mediated mitochondrial dysfunction, consistent with recent studies [48].

Considering that the mitochondrion is the main site of FA ß-oxidation and the key role of FA ß-oxidation on hepatic lipid metabolism is homeostasis, we next explored the role of SIRT3-mediated UPRmt on mitochondrial FA ß-oxidation and lipid deposition. First, similar to other studies [49], we found that HFD/FA caused the disorders for mitochondrial FA ß-oxidation, but choline obviously improved mitochondrial FA ß-oxidation. Then, our results reveled the interaction between mtHSP70 and ACOX1. Since ACOX1 is the first and rate-limiting enzyme in FA β-oxidation [50], this result provided the first evidence of the direct links between UPRmt and FA ß-oxidation. Furthermore, in vitro experimentation found that this interaction was regulated by FA/choline treatment: FA promoted this interaction, but choline has attenuated effects. These suggested the involvement of FA/choline in mtHSP70–ACOX1 interaction. On the other hand, SIRT3 knockdown resulted in the downregulation of ACOX1, implying the regulatory role of SIRT3 in ACOX1. Then, SIRT3 knockdown led to the decreased mitochondrial PA β-oxidation efficiency and acetyl-CoA content, which further confirmed the regulatory role of SIRT3 on mitochondrial FA β-oxidation. Similarly, the study found that SIRT3 knockdown caused reduced mitochondrial oxidation in cultured myoblasts of mice in vitro [51]. However, considering the critical role of SIRT3 in the mitochondrial metabolism, the enhanced mtHSP70–ACOX1 interaction and descending β-oxidation may also be attributed to reduced SIRT3 function and its subsequent effect after FA treatment, not just the decreased SIRT3 expression. Importantly, we also demonstrated that SIRT3 knockdown caused excessive lipid droplet accumulation and also led to the loss of choline-activated ameliorating effects. This indicates the negative regulatory role of SIRT3 on hepatic lipid accumulation. Similarly, Sheng et al. [52] suggested that the overexpression of SIRT3 inhibits lipid accumulation in macrophages in vitro.

In conclusion, the present study clearly demonstrated that HFD induced mitochondrial dysfunction via UPRmt, which was regulated SIRT3-mediated mtHSP70 acetylation. Meanwhile, UPRmt regulated FA ß-oxidation though the interaction of mtHSP70 with ACOX1. Furthermore, choline showed alleviating effects on the above-mentioned process via SIRT3. The detail mechanism have been shown in Figure 7.

## Figures and Tables

**Figure 1 ijms-23-04204-f001:**
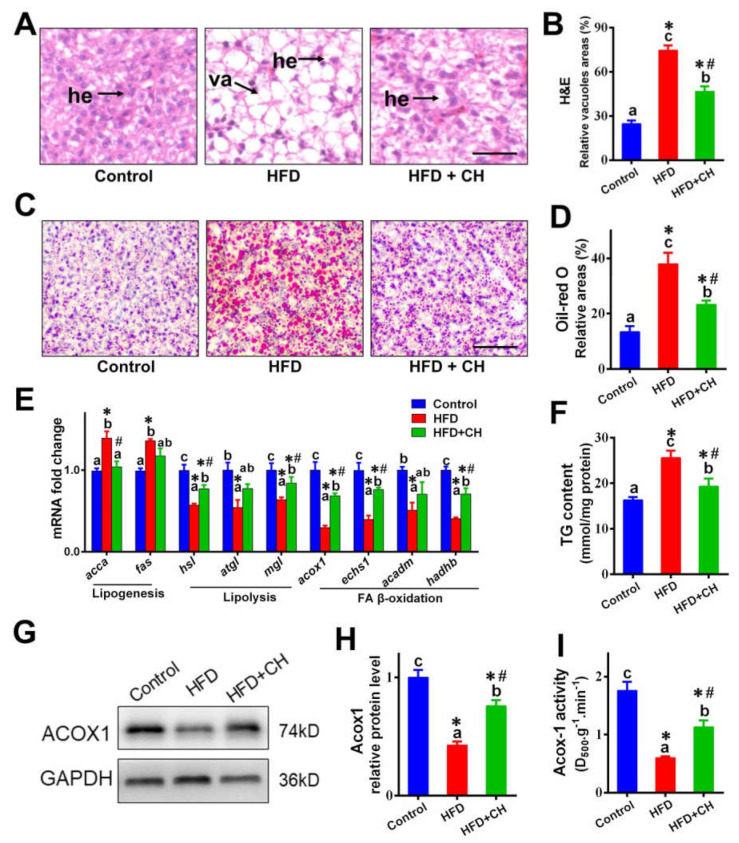
Dietary choline attenuated high-fat diet (HFD)-induced hepatic lipid dysregulation of yellow catfish. (**A**) Representative images of liver H&E staining. Scale bar, 30 μm, hepatocytes (he), vacuoles (va). (**B**) Relative areas for hepatic vacuoles in H&E staining. (**C**) Representative images of liver Oil Red O staining. Scale bar, 30 μm. (**D**) Relative areas for LDs in Oil Red O staining. (**E**) The mRNA levels of the genes related to hepatic lipid metabolism. (**F**) Hepatic TG content. (**G**,**H**) Western blot analysis and quantification analysis of ACOX1. (**I**) Activities of ACOX1 enzymes. Data are mean ± SEM (*n* = 3 replicate tanks), different letters indicate significant differences among groups, asterisks * indicate significant differences between control and HFD/or HFD + CH, and pound # indicates significant differences between HFD and HFD + CH (*p* ≤ 0.05).

**Figure 2 ijms-23-04204-f002:**
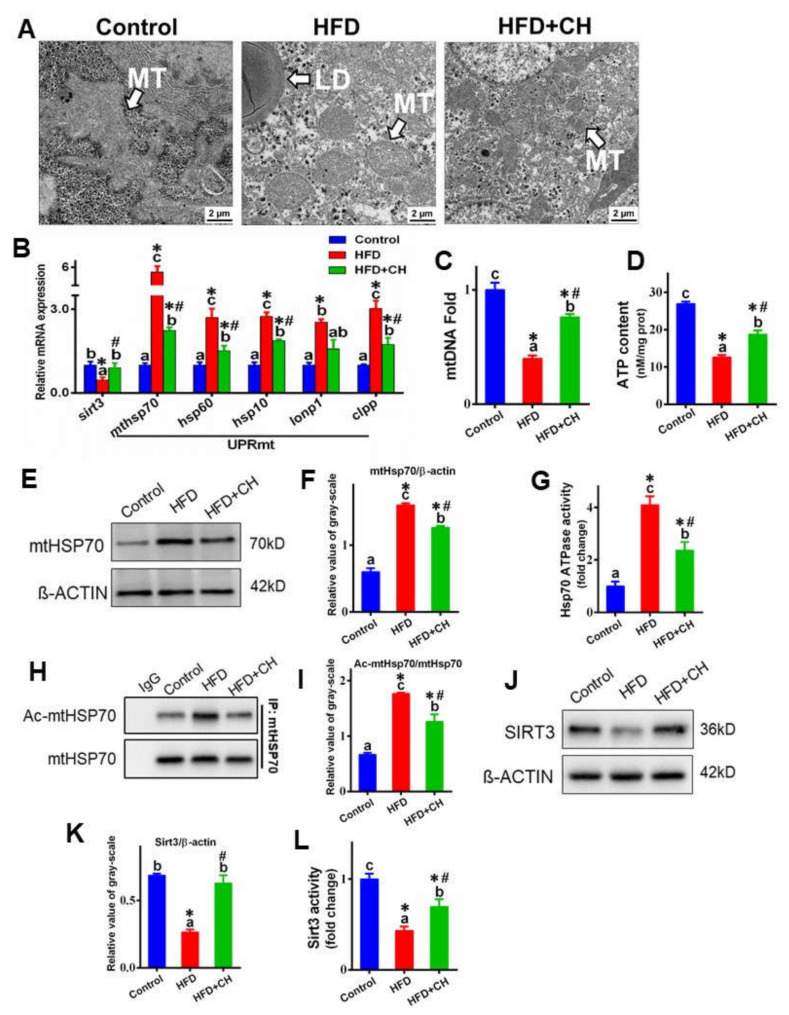
The involvement of SIRT3-mediated mtHSP70 acetylation in dietary choline attenuating HFD-induced mitochondrial unfolded protein response (UPRmt) in the livers of yellow catfish. (**A**) TEM structures of the liver, mitochondrion (MT), lipid droplet (LD); scale bars, 2 μm. (**B**) The mRNA levels of the genes related to SIRT3 and UPRmt. (**C**) The mtDNA fold. (**D**) ATP content. (**E**,**F**) Western blot analysis and quantification analysis of mtHSP70. (**G**) HSP70-ATPase activity. (**H**,**I**) Western blot analysis and quantification analysis of mtHSP70 acetylation. (**J**,**K**) Western blot analysis and quantification analysis of SIRT3. (**L**) SIRT3 deacetylase activity. Data are mean ± SEM (*n* = 3 replicate tanks). Different letters indicate significant differences among groups, asterisks * indicate significant differences between control and HFD/or HFD + CH, and pound # indicates significant differences between HFD and HFD + CH (*p* ≤ 0.05).

**Figure 3 ijms-23-04204-f003:**
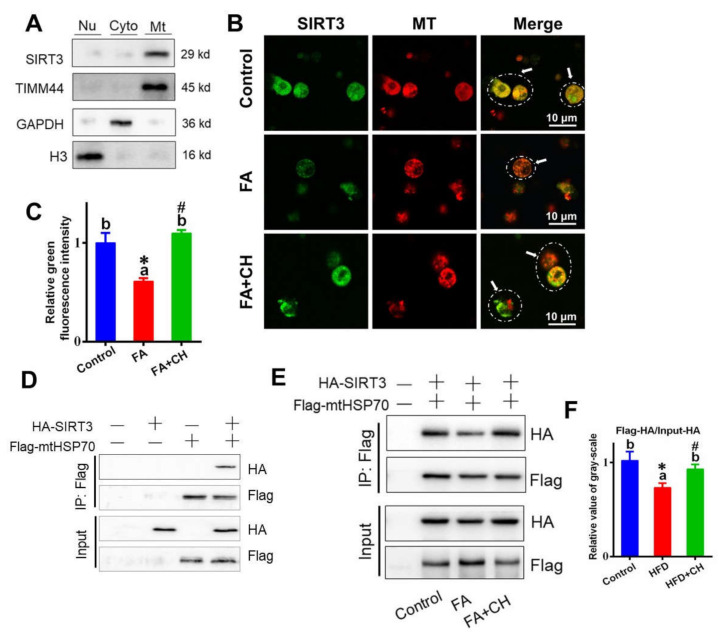
Choline improved the subcellular localization of SIRT3 into the mitochondria (MT) and alleviated the FA-reduced interaction of SIRT3 with mtHSP70 in vitro. (**A**) Western blot-analyzed subcellular localization of SIRT3 in hepatocytes of yellow catfish. Timm, mitochondrion marker; GAPDH, cytoplasm marker; H3, nucleus marker. (**B**) Co-localization analysis of SIRT3 (green) and mitochondria (red) (bars = 10 μm) in hepatocytes of yellow catfish. Arrows indicate the regions of co-localization of SIRT3 and mitochondria. (**C**) Quantitative analysis for relative green intensity of fluorescence in B. (**D**,**E**) Interaction of mtHSP70 with SIRT3 without/with FA and choline incubation. FLAG-tag mtHSP70 and HA-tag SIRT3 were transfected into the HepG2 cell line. The interaction between mtHSP70 and SIRT3 was determined with immunoprecipitation and Western blot analysis in HepG2 cell whole cell homogenates. (**F**) Relative value of gray-scale analysis for panel E. Values are mean ± SEM (*n* = 3 independent biological experiments); different letters indicate significant differences among groups; asterisks * indicate significant differences between control and HFD/or HFD + CH, and pound # indicates significant differences between HFD and HFD + CH (*p* ≤ 0.05).

**Figure 4 ijms-23-04204-f004:**
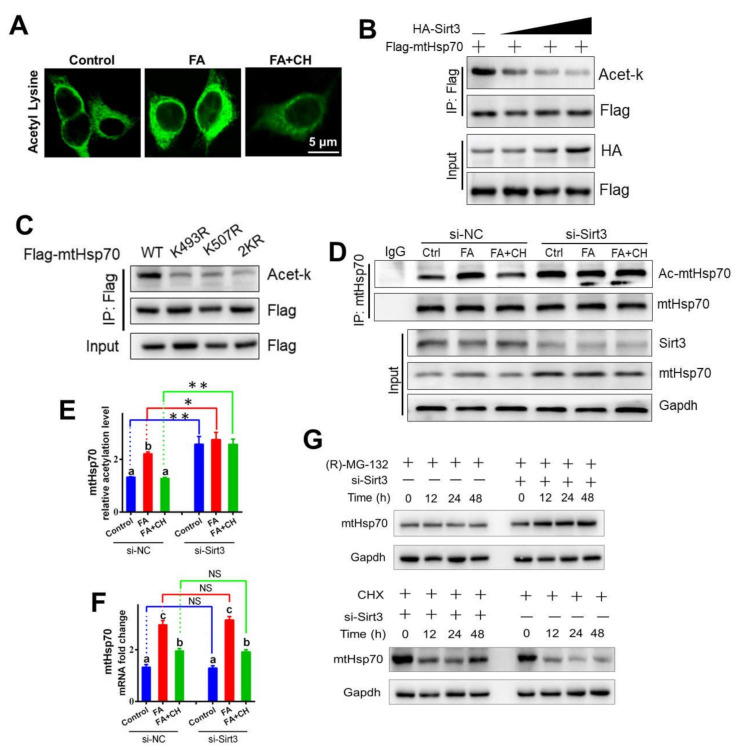
Choline-inhibited FA increased the stability of mtHSP70 via SIRT3-mediated deacetylation in vitro. (**A**) Immunofluorescent analysis of acetylation after the anti-acetyl lysine antibody (green) in hepatocytes of yellow catfish. Bars represent 5 μm. (**B**) Overexpression of SIRT3 decreased the acetylation level of mtHSP70. Different doses of HA-tag SIRT3 and FLAG-tag mtHSP70 were transfected into the HepG2 cell line. The acetylation level of mtHSP70 was determined with immunoprecipitation and Western blot analysis. (**C**) Mutations of K493R and K507R decreased the mtHSP70 acetylation level. The acetylation of ectopically expressed WT, K493R, K507R, and 2KR was analyzed. (**D**) The hepatocytes of yellow catfish were transfected with SIRT3 siRNA, and then the SIRT3 and mtHSP70 protein levels were detected using Western blot analysis. The acetylation level of mtHSP70 was determined using immunoprecipitation and Western blot analysis. (**E**) The quantitative analysis acetylation level of mtHSP70 in D. (**F**) The mRNA levels of mtHSP70 in the hepatocytes of yellow catfish. (**G**) The hepatocytes of yellow catfish were transfected with SIRT3 siRNA and also treated with (H)-MG-132 or CHX for 48 h, and then the mtHSP70 protein levels were detected using Western blot analysis. Values are mean ± SEM (*n* = 3 independent biological experiments); different letters indicate significant differences among same si-NC or si-SIRT3 groups (*p* ≤ 0.05); asterisks indicate significant differences between si-NC and si-SIRT3 groups (* *p* ≤ 0.05; ** *p* ≤ 0.01).

**Figure 5 ijms-23-04204-f005:**
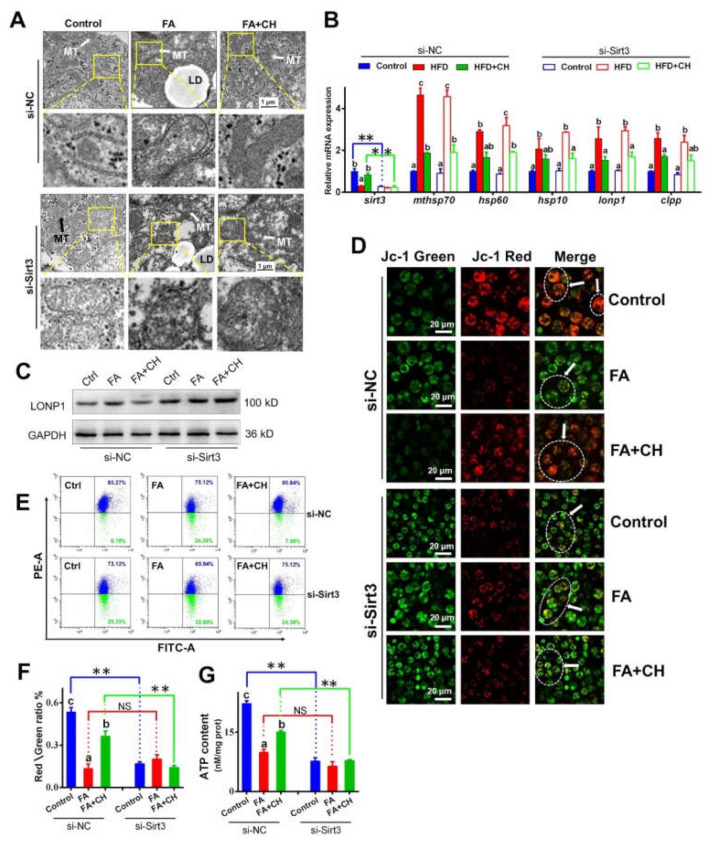
SIRT3/mtHSP70-mediated UPRmt is necessary for choline-ameliorating FA-induced mitochondrial dysfunction in the hepatocytes of yellow catfish. (**A**) TEM structures of the hepatocytes incubated by FA and transfected with SIRT3 siRNA, MT, lipid droplet (LD); scale bars, 1 μm. (**B**) The mRNA levels of the genes related to SIRT3 and UPRmt. (**C**) Western blot analysis of LONP1. (**D**,**E**) Images of hepatocytes stained by JC-1 output by fluorescence scanning confocal microscopy (D, bars = 20 μm) and flow cytometry analysis (**E**) under FA incubation and transfected with SIRT3 siRNA. Representative images of confocal microscopy were marked with arrows. (**F**) Quantitative analysis for the relative ratio of red/green fluorescence intensity. (**G**) The ATP content. Values are mean ± SEM (*n* = 3 independent biological experiments); different letters indicate significant differences among same si-NC or si-SIRT3 groups (*p* ≤ 0.05); asterisks indicate significant differences between si-NC and si- SIRT3 groups (* *p* ≤ 0.05; ** *p* ≤ 0.01).

**Figure 6 ijms-23-04204-f006:**
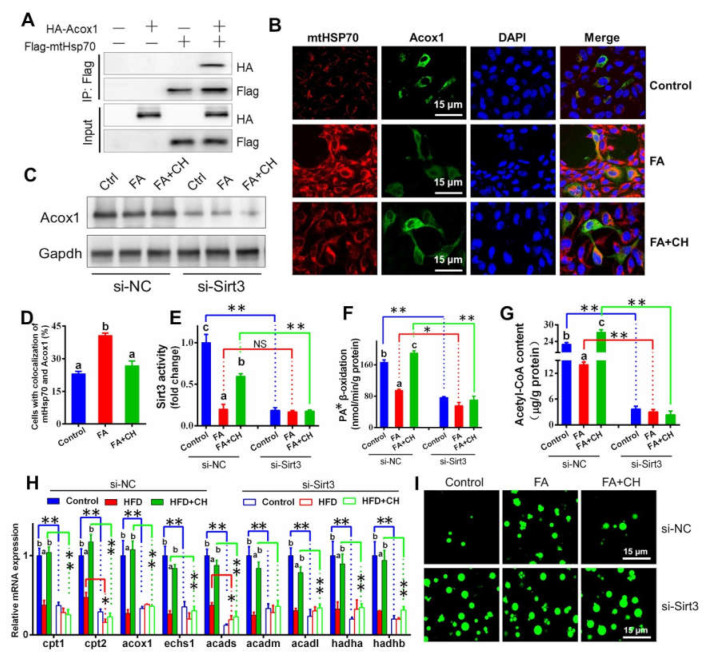
Choline alleviating FA-induced hepatic lipid dysregulation via SIRT3-mediated interaction between mtHSP70 and ACOX1 in the hepatocytes of yellow catfish. (**A**) Interaction of mtHSP70 with ACOX1. FLAG-tag mtHSP70 and HA-tag ACOX1 were transfected into the HepG2 cell line. The interaction between mtHSP70 and SIRT3 was determined using immunoprecipitation and Western blot analysis. (**B**) Co-localization analysis of mtHSP70 (red) and ACOX1 (green) in the hepatocytes of yellow catfish (blue for nucleus) (bars = 15 μm). (**C**) Western blot analysis of ACOX1. (**D**) Quantitative analysis for co-localization between mtHSP70 and ACOX1 in B. (**E**) SIRT3 deacetylase activity. (**F**) Mitochondrial palmitic acid β-oxidation efficiency. (**G**) Acetyl-CoA content. (**H**) The mRNA levels of the genes related to mitochondrial FA β-oxidation. (**I**) Representative confocal microscopic image of lipid droplets (green) hepatocytes of yellow catfish (bar = 15 μm). Values are mean ± SEM (*n* = 3 independent biological experiments); different letters indicate significant differences among same si-NC or si-SIRT3 groups (*p* ≤ 0.05); asterisks indicate significant differences between si-NC and si-SIRT3 groups (* *p* ≤ 0.05; ** *p* ≤ 0.01).

**Figure 7 ijms-23-04204-f007:**
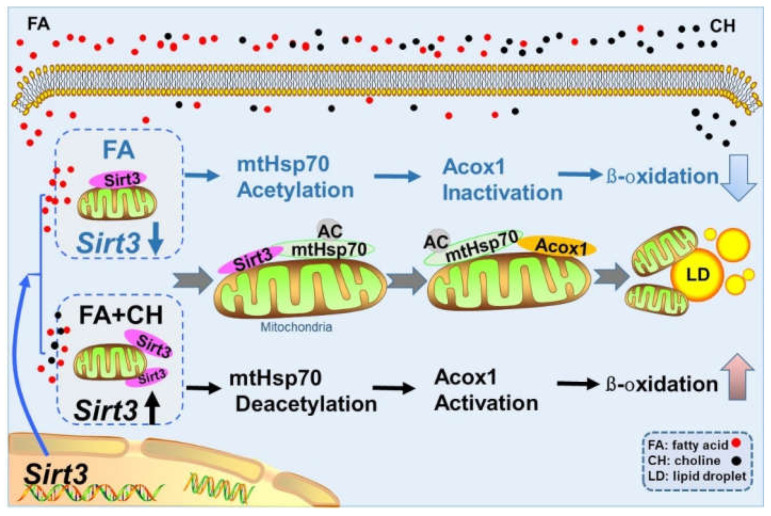
Graphical conclusions for the mechanism of dietary choline alleviating HFD-induced hepatic lipid dysregulation via UPRmt modulated by SIRT3-mediated mtHSP70 deacetylation.

## Data Availability

Not applicable.

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
