# Peer review of "Dietary Choline Alleviates High-Fat Diet-Induced Hepatic Lipid Dysregulation via UPRmt Modulated by SIRT3-Mediated mtHSP70 Deacetylation"

_ijms, 2022, doi:10.3390/ijms23084204_

Round 1
Reviewer 1 Report
This manuscript by Song et al. demonstrates that dietary choline alleviates HFD-induced hepatic lipid dysregulation via UPRmt modulated by Sirt3-mediated mtHsp70 deacetylation. Overall the results presented are interesting. Here are my comments:
- There are grammatical errors throughout the manuscript. e.g. line52, line78, and so on.
- In the introduction, line67 doesn't have a reference. Please discuss mitochondrial Hsp70 post-translational modifications and refer to PMID:32518165.
- In Figure1,2,3 in the bar graphs please show the significance.
- In Figures 3B & 5D, please mark with arrows the specific regions.
- The authors show that acetylation mutants result in decreased mtHsp70 acetylation. Do the authors get the same results when they use a deacetylase inhibitor?
- What about acetylation mimic mutants of mt HSP70 sites K493, and K507? Do these mutations mimic wild type?
- Can the authors decipher what is the order of mtHsp70 deacetylation? Is it sequential or simultaneous?
- What are the acetylases that add on these acetylations?
Author Response
Responses to Reviewers’ Comments
Title: Dietary Choline Alleviates High Fat Diet-Induced Hepatic Lipid Dysregulation via UPRmt Modulated by Sirt3-Mediated mtHsp70 Deacetylation
Ms. Ref. No.: ijms-1607618
Corresponding Author: Prof. Yu-Feng Song
Date: 2022/14/03
Dear Dr. the Reviewer,
Thank you and anonymous reviewers very much for your enormous contributions to our manuscript and for your constructive comments. These comments are very pertinent and helpful for improving the quality of our manuscript, and also helpful for our research in the future. We have strictly revised our manuscript based on these valuable suggestions, and, as a matter of fact, we have done our best to revise it. We hope that you are satisfactory to our revisions and expect your positive response. Still, if you encounter any questions, please do not hesitate to contact us. Thanks in advance.
Here, we listed our responses, on a point-to-point basis, to your important comments.
Reviewer # 1:
This manuscript by Song et al. demonstrates that dietary choline alleviates HFD-induced hepatic lipid dysregulation via UPRmt modulated by Sirt3-mediated mtHsp70 deacetylation. Overall the results presented are interesting. Here are my comments:
Comment 1: There are grammatical errors throughout the manuscript. e.g. line52, line78, and so on.
Response 1: Thanks for your valuable comments. Based on your comments, we have checked and revised all grammatical errors throughout the manuscript. For the detail, please see the text.
Comment 2: In the introduction, line67 doesn't have a reference. Please discuss mitochondrial Hsp70 post-translational modifications and refer to PMID:32518165.
Response 2: Thanks for your valuable comments. We have discussed the mitochondrial Hsp70 post-translational modifications based on the PMID:32518165. For the detail, please see the text.
Comment 3: In Figure1,2,3 in the bar graphs please show the significance.
Response 3: Thanks for your valuable comments. We have add the significance in Figure1,2,3 in the bar graphs. For the detail, please see the text.
Comment 4: In Figures 3B & 5D, please mark with arrows the specific regions.
Response 4: We have marked the specific regions with arrows based on your comments. For the detail, please see the text.
Comment 5: The authors show that acetylation mutants result in decreased mtHsp70 acetylation. Do the authors get the same results when they use a deacetylase inhibitor?
Response 5: Thanks for your valuable comments. Sorry, in this present study, we didn't use deacetylase inhibitor to make sure the effect of acetylation mutants. However, in our present study, we determined acetylation mutants based on not only the protein sequence alignment of mtHsp70 homologues, but also liquid chromatography-mass spectrometry analysis for mtHsp70. All these could make sure the result that acetylation mutants decreased mtHsp70 acetylation.
Comment 6: What about acetylation mimic mutants of mtHSP70 sites K493, and K507? Do these mutations mimic wild type?
Response 6: Thanks for your valuable comments. In this present study, we lose the data about acetylation mimic mutants (K-Q) for sites K493, and K507. But in our present study, the result of WT (Fig.4 C), to a certain extent, could mimic wild type of acetylation sites. On the other hand, in our previous study, we did use the same way to test the acetylation sites (Wei et al., 2021). References:Wei XL, Hogstrand C, Chen GH, Lv WH, Song YF, Xu YC, Luo Z. Zn Induces Lipophagy via the Deacetylation of Beclin1 and Alleviates Cu-Induced Lipotoxicity at Their Environmentally Relevant Concentrations. Environ Sci Technol. 2021 20;55(8):4943-4953.
Comment 7: Can the authors decipher what is the order of mtHsp70 deacetylation? Is it sequential or simultaneous?
Response 7: Thanks for your valuable comments. Here, honestly, from our present result it is difficult to decipher the order of mtHsp70 deacetylation. mtHsp70 deacetylation were involved in complex system of deacetylases. Although we can make sure the main deacetylases for mtHsp70 was Sirt3, we still cannot eliminate interferences from other deacetylases. So we didn't get the enough data for the order of mtHsp70 deacetylation. However, we will focus on this interesting part, and hope can get the conclusion in our future research.
Comment 8: What are the acetylases that add on these acetylations?Response 8: In this present study, we only focus on the deacetylases for mtHsp70, and conclude that Sirt3 regulated mtHsp70 deacetylation in a dose-dependent manner. But we didn't test which one of acetylases responsible for mtHsp70 acetylations.
Reviewer 2 Report
The manuscript by Yu-Feng Song et al. features an in vivo and in vitro study dealing with the mechanism underlying lipid metabolism dysregulation in the liver and the protective effects of choline. The authors conclude that the mechanism is related to mitochondrial unfolded protein response stress via changes in sirtuin 3 and Hsp70, among others. My comments are as follows.
MAJOR COMMENTS
- The manuscript has numerous defects in the presentation regarding not only blatant English errors but also indiscriminate use of abbreviations without previous definition. Examples of the former just in the abstract: ‘study taken’ (L14), ‘as grasp to’ (L15), ‘cells line’ (L18), ‘in control solution or fatty acids’ (L19), and so on. In particular, the expression ‘in vivo trail’ had me puzzled for quite some time. Please note it is not only specific words or constructions here and there, the meaning is lost frequently (as in lines 68-73, for instance). Of the latter: UPRmt, HFD, FA (just in the abstract). In addition, from a strictly scientific point of view, the writing is confusing and unnecessarily complex. All this requires an extensive revision.
- Several figures include a heatmap corresponding to RT-PCR data. This is incorrect, as it gives no idea of variability and is visually less possible to gauge for the reader. Instead, please provide standard graphs.
- The use of a 3 group design is debatable. It would be nice to see whether choline has effects in basal conditions, particularly since choline deficiency has been long known to induce steatosis.
- The authors do not justify the use of two different in vitro models. The reason is perhaps obvious but nevertheless it should not be something to wonder about.
- In the in vitro experiments the control is said to have ‘no extra addition’. How are fatty acids added (i.e. any vehicle)?
- Section 2.2.5.: I did not understand this part at all, it refers to mtDNA measured in plasma (!).
- The authors use several antibodies for which fish reactivity is not characterized. This includes ab217319 (human), ab2799 (various species including rat and mouse), ab184032 (human, rat, mouse). This may be featured in previous papers, if so this should be addressed and referenced. If not, it is required in my opinion to do the necessary checks.
- The description of both methods and data regarding flow cytometry and mitochondrial membrane potential is inadequate in my opinion. Also, in general, there is no description of what the samples are (in all measurements), i.e. where are things measured (see also point 10). This makes interpretation quite difficult.
- Fig. 1: in panel A, do the authors suggest that vacuoles are not located in hepatocytes? Panel B: revise Y axis label.
- Fig. 2: Apart from the issue with mtDNA data mentioned above, panel D has no quantification (also panel G). Data in panels E and F are difficult to interpret, since no details are provided for the technique in the first case and for the samples analyzed in the second (the same happens with panel H, which is also not very convincing as there appears to be an artifact in the blot). Please note that these data suggest but not imply (L298) Sirt3 is causing mtHsp70 deacetylation. In fact, based on what is shown in panel G choline produces no change in mtHsp70 acetylation status.
- Fig. 3: fractioning in panel A is not described. In panel E there is a lower Sirt3 signal in the mtHsp70 IP, but this may be due to the decreased levels (HA signal in input and Fig. 2H).
- Fig. 4: I can make no sense of panel A I am afraid. This shows the impact of FA on overall Lys acetylation, however there is no red signal at all in these images. Assuming it is green, there is indeed increased acetylation with FA, which is completely normalized with choline, but there is no point in doing the merge, is there? Mitochondrial rather than nuclear staining would be expected here. Even though this is the effect on Lys acetylation, Fig. 2G shows no effect of choline on mtHsp70 acetylation, as mentioned above.
Why is IP required in panel B?
In turn, the siRNA experiment does show the effect on of Sirt3 on mtHsp70 acetylation (please note that the text is wrong in L351), where basically downregulation of Sirt3 by siRNA has a similar effect to FA. However, the levels of Sirt3 are barely altered in the latter. This suggests that the effect of FA is not due to lower Sirt3 expression. Interestingly, panel G shows that ATPase activity is further increased by Sirt3 knockdown (although levels are unchanged). In panel I Sirt3 knockdown is said to reduce ubiquitination, however I do not see that, at least in the presence of PR-619, nor do I see a significant change in acetylation.
- Fig. 5: I am afraid I cannot see much in panel A, this should be much improved. Jc-1 data are similar to previous observations in that Sirt3 knockdown has effects akin to those of FA. As commented above, the methods requires explanation.
- Fig. 6: again, the text (L413) is inconsistent with the data. FA enhance mtHsp70 – Acox interaction. Panel C shows that si-Sirt3 results in greatly downregulated Acox1 levels, but if we look at panel E, FA causes a comparable reduction of Sirt3 activity but no change at all in Acox1. This is consistent with FA acting by Sirt3 inhibition (rather than downregulation of expression). On the other hand, the si-Sirt3 data confirm that Sirt3 is required for beta-oxidation rather than prove that Sirt3 mediates the choline effect.
- The Discussion is too long. In particular, the first paragraph is expendable. The remainder would be better if substantially condensed. I must say I see again the authors present the data wrongly. As far as I can tell HFD upregulates mtHsp70 and increases acetylation, not the other way around, via Sirt3 exit to the cytoplasm. Increased mtHsp70 seems to inhibit Acox1. However, the authors fail to show choline affecting either. Perhaps Sirt3 is working via other mechanisms. Because Sirt3 knockdown has such a great effect by itself not that much can be deduced from these experiments in my opinion. I was about to suggest adding a diagram showing the findings of the study and highlighting what remains to be known; then I found such a diagram was actually provided, but only at the very end of supplementary material document, and it is not referred to in the text. This diagram is consistent with my interpretation of the data, with some differences. FA are displayed as uptaken by an apparent channel, this should be modified. I was surprised also to see that there is a second file labelled as ‘non published’ which appears to be identical to the supplementary data file, except for this diagram.
Also, HFD upregulates the expression of lipogenic genes, this aspect is pretty much ignored in the rest of the article.
The authors are unclear regarding the novelty of mtUPR (I guess this is preferable to UPRmt) in connection with HFD.
- It is advisable to produce the original blots.
MINOR COMMENTS
- Protein symbols should be in capital letters, gene symbols in italics (but not as written in L155 onwards either).
- L74: lipotropic? L75: NRC, 2011? L83: duplicated twice?
- L83 on: this explanation is confusing, the authors appear to hint that the extra duplication in fish is relevant for susceptibility to a high fat diet (HFD), but that would make fish a less relevant model for other vertebrates, and HFD do produce steatosis generally. So the point of this part is questionable.
- L108: it is best to provide the actual reference.
- L121: growth rate and weight gain are calculated in bulk? Also some more detail about these measurements would be advisable. Why only 9 out of 75 fish for structural analysis? What is the size of a 3 g fish and how it influences analysis for biochemical parameters? What do the authors mean exactly in L140? Also please use names consistently (i.e. scientific and normal name of the species).
- L127: these fishes as twice as big, why? What is the life span of cultured hepatocytes? What are OA and PA?
- L157: how stable were those genes?
- L169: if a kit for measuring deacetylase activity is used, this is what ought to be reported, not Sirt3 activity. L171: please describe assay briefly. L172: what do the authors mean by ‘approximate single turnover measurements’?
- L195: provide details of equipment used.
- What was the efficiency of siRNA transfection?
- L286: it is decreased; L287: but?
- Fig. 4C: what do the authors make of the 2KR signal being so similar to that of the single mutants?
- Revise labeling in Y axis, panel 6F.
- Please avoid the expression ‘upregulation of acetylation’.
Author Response
Dear Dr. the Reviewer,
Thank you and anonymous reviewers very much for your enormous contributions to our manuscript and for your constructive comments. These comments are very pertinent and helpful for improving the quality of our manuscript, and also helpful for our research in the future. We have strictly revised our manuscript based on these valuable suggestions, and, as a matter of fact, we have done our best to revise it. We hope that you are satisfactory to our revisions and expect your positive response. Still, if you encounter any questions, please do not hesitate to contact us. Thanks in advance.
Here, we listed our responses, on a point-to-point basis, to your important comments.
Reviewer # 2:
The manuscript by Yu-Feng Song et al. features an in vivo and in vitro study dealing with the mechanism underlying lipid metabolism dysregulation in the liver and the protective effects of choline. The authors conclude that the mechanism is related to mitochondrial unfolded protein response stress via changes in sirtuin 3 and Hsp70, among others. My comments are as follows.
Major comments:
Comment 1: The manuscript has numerous defects in the presentation regarding not only blatant English errors but also indiscriminate use of abbreviations without previous definition. Examples of the former just in the abstract: ‘study taken’ (L14), ‘as grasp to’ (L15), ‘cells line’ (L18), ‘in control solution or fatty acids’ (L19), and so on. In particular, the expression ‘in vivo trail’ had me puzzled for quite some time. Please note it is not only specific words or constructions here and there, the meaning is lost frequently (as in lines 68-73, for instance). Of the latter: UPRmt, HFD, FA (just in the abstract). In addition, from a strictly scientific point of view, the writing is confusing and unnecessarily complex. All this requires an extensive revision.
Response 1: Sorry about so many errors in our article, that make you puzzled. Here, based on your comments, we have made an extensive revision and deleted a lot of unnecessarily writing. For the detail, please see the text. Thanks for your detail comments.
Comment 2: Several figures include a heatmap corresponding to RT-PCR data. This is incorrect, as it gives no idea of variability and is visually less possible to gauge for the reader. Instead, please provide standard graphs.
Response 2: We have provide standard graphs for Fig. 1E, Fig. 2B, Fig. 5B and Fig. 6H instead of heatmap. For the detail, please see the text.
Comment 3: The use of a 3 group design is debatable. It would be nice to see whether choline has effects in basal conditions, particularly since choline deficiency has been long known to induce steatosis.
Response 3: Thanks for your valuable comments. First, we are totally agree with you about the addition of choline basal condition. But this present study is a continuation and extension of our previous studies. In our lab, the effect of dietary choline levels on growth performance and lipid metabolism in yellow catfish have been confirmed (Luo et al., 2016). On the other hand, in our present study, the Contorl and HFD groups also added choline (see the Supplementary Table 1) based on our previous study (Luo et al., 2016), which will make sure no choline deficiency for yellow catfish. Lastly, choline is known as lipid-lowering agent for fish and have been widely used in aquaculture. So the similar condition (no choline basal condition) was also been researched in other fish (Li et al., 2014).
References:Li JY, Zhang D, Xu WN, Jiang GZ, Zhang CN, Li XF. Effects of dietary choline supplementation on growth performance and hepatic lipid transport in blunt snout bream (megalobrama amblycephala) fed high-fat diets. Aquaculture, 2014. 434, 340-347.Luo Z, Wei CC, Ye HM, Zhao HP, Song YF, Wu K. Effect of dietary choline levels on growth performance, lipid deposition and metabolism in juvenile yellow catfish Pelteobagrus fulvidraco. Comp Biochem Physiol B Biochem Mol Biol. 2016. 202:1-7. doi: 10.1016/j.cbpb.2016.07.005.
Comment 4: The authors do not justify the use of two different in vitro models. The reason is perhaps obvious but nevertheless it should not be something to wonder about.
Response 4: Thanks for your valuable comments. We used two different in vitro models, because it is quite difficult to do immunoprecipitation for hepatocytes of P. fulvidraco. So we have to use HepG2 cell line to verify these interaction. The similar way for testing interaction in yellow catfish was also been determined (Wei et al., 2021). At the same time, we also explain this in Materials and methods section. For the detail, please see the text.
References:Wei X, Hogstrand C, Chen G, Lv W, Song Y, Xu Y, Luo Z. Zn Induces Lipophagy via the Deacetylation of Beclin1 and Alleviates Cu-Induced Lipotoxicity at Their Environmentally Relevant Concentrations. Environ Sci Technol. 2021. 20;55(8):4943-4953. doi: 10.1021/acs.est.0c08609.
Comment 5: In the in vitro experiments the control is said to have ‘no extra addition’. How are fatty acids added (i.e. any vehicle)?
Response 5: “no extra addition” meaning no extra fatty acid addition. In our present study, fatty acid was added by DMSO. They were dissolved in DMSO before addition. The way of fatty acid addition was according to our previous studies (Song et al., 2020). At the same time, we explain this information in Materials and methods section. For the detail, please see the text.
References:
Song YF, Hogstrand C, Ling SC, Chen GH, Luo Z. Creb-Pgc1α pathway modulates the interaction between lipid droplets and mitochondria and influences high fat diet-induced changes of lipid metabolism in the liver and isolated hepatocytes of yellow catfish. J Nutr Biochem. 2020. doi: 10.1016/j.jnutbio.2020.108364.
Comment 6: Section 2.2.5.: I did not understand this part at all, it refers to mtDNA measured in plasma (!).
Response 6: Sorry here make you puzzle again. mtDNA measured by DNA copy number in cytoplasm, not in plasma. The determination was total according our previous study (Song et al., 2020), and we have detail the methods of mtDNA measurement. For the detail, please see the text.
References:
Song YF, Hogstrand C, Ling SC, Chen GH, Luo Z. Creb-Pgc1α pathway modulates the interaction between lipid droplets and mitochondria and influences high fat diet-induced changes of lipid metabolism in the liver and isolated hepatocytes of yellow catfish. J Nutr Biochem. 2020. doi: 10.1016/j.jnutbio.2020.108364.
Comment 7: The authors use several antibodies for which fish reactivity is not characterized. This includes ab217319 (human), ab2799 (various species including rat and mouse), ab184032 (human, rat, mouse). This may be featured in previous papers, if so this should be addressed and referenced. If not, it is required in my opinion to do the necessary checks.
Response 7: Thanks for your valuable comments. In our lab, due to there are no specific commercial antibody for yellow catfish. So we have used antibodies from other various species, including human, rat, mouse for many years. And many studies shown it is feasible (Wei et al., 2018; Chen et al., 2019; Ling et al., 2019; Song et al., 2020; Zhao et al., 2020; Wei et al., 2021). As for these antibodies in our present also been used in our lab for some previous study (Wei et al., 2021) or unpublished research. Based on your comments, we have added some references for this section. For the detail, please see the text.
References:
Chen G, Wu K, Zhao T, et al. miR-144 Mediates High Fat–Induced Changes of Cholesterol Metabolism via Direct Regulation of C/EBPα in the Liver and Isolated Hepatocytes of Yellow Catfish. J Nutr, 2019. DOI: 10.1093/jn/nxz282.
Ling S C, Wu K, Zhang D G, et al. Endoplasmic Reticulum Stress–Mediated Autophagy and Apoptosis Alleviate Dietary Fat–Induced Triglyceride Accumulation in the Intestine and in Isolated Intestinal Epithelial Cells of Yellow Catfish. J Nutr, 2019. doi: org/10.1093/jn/nxz135.
Song YF, Hogstrand C, Ling SC, Chen GH, Luo Z. Creb-Pgc1α pathway modulates the interaction between lipid droplets and mitochondria and influences high fat diet-induced changes of lipid metabolism in the liver and isolated hepatocytes of yellow catfish. J Nutr Biochem. 2020. doi: 10.1016/j.jnutbio.2020.108364.
Wei CC, Luo Z, Hogstrand C, Xu YH, Wu LX, Chen GH, Pan YX, Song YF. Zinc reduces hepatic lipid deposition and activates lipophagy via Zn2+/MTF-1/PPARα and Ca2+/CaMKKβ/AMPK pathways. FASEB J. 2018. 18:fj201800463. doi: 10.1096/fj.201800463.
Wei X, Hogstrand C, Chen G, Lv W, Song Y, Xu Y, Luo Z. Zn Induces Lipophagy via the Deacetylation of Beclin1 and Alleviates Cu-Induced Lipotoxicity at Their Environmentally Relevant Concentrations. Environ Sci Technol. 2021 Apr 20;55(8):4943-4953. doi: 10.1021/acs.est.0c08609.
Zhao T, Wu K, Hogstrand C, Xu YH, Chen GH, Wei CC, Luo Z. Lipophagy mediated carbohydrate-induced changes of lipid metabolism via oxidative stress, endoplasmic reticulum (ER) stress and ChREBP/PPARγ pathways. Cell Mol Life Sci. 2020. 77(10):1987-2003. doi: 10.1007/s00018-019-03263-6.
Comment 8: The description of both methods and data regarding flow cytometry and mitochondrial membrane potential is inadequate in my opinion. Also, in general, there is no description of what the samples are (in all measurements), i.e. where are things measured (see also point 10). This makes interpretation quite difficult.
Response 8: Thanks for your valuable comments. We have added more detail about measurement method of flow cytometry and mitochondrial membrane potential according our previous study (Song et al., 2018, 2020). For the detail, please see the text.
References:
Song YF, Gao Y, Hogstrand C, Li DD, Pan YX, Luo Z. Upstream regulators of apoptosis mediates methionine-induced changes of lipid metabolism. Cell Signal. 2018. 51:176-190. doi: 10.1016/j.cellsig.2018.08.005.
Song YF, Hogstrand C, Ling SC, Chen GH, Luo Z. Creb-Pgc1α pathway modulates the interaction between lipid droplets and mitochondria and influences high fat diet-induced changes of lipid metabolism in the liver and isolated hepatocytes of yellow catfish. J Nutr Biochem. 2020. doi: 10.1016/j.jnutbio.2020.108364.
Comment 9: Fig. 1: in panel A, do the authors suggest that vacuoles are not located in hepatocytes? Panel B: revise Y axis label.
Response 9: Yellow catfish exhibits some hepatic lipid accumulation even after feedin commercial feed. This have been proved in our previous studies, and also shown in H&E (Wei et al., 2018; Chen et al., 2019). So in Fig. 1 A there are some vacuoles, but it was significantly less than HFD. On the other hand, we have revise Y axis label in Fig. 1 B. For the detail, please see the text.
References:
Chen G, Wu K, Zhao T, et al. miR-144 Mediates High Fat–Induced Changes of Cholesterol Metabolism via Direct Regulation of C/EBPα in the Liver and Isolated Hepatocytes of Yellow Catfish. J Nutr, 2019. DOI: 10.1093/jn/nxz282.
Wei CC, Luo Z, Hogstrand C, Xu YH, Wu LX, Chen GH, Pan YX, Song YF. Zinc reduces hepatic lipid deposition and activates lipophagy via Zn2+/MTF-1/PPARα and Ca2+/CaMKKβ/AMPK pathways. FASEB J. 2018. 18:fj201800463. doi: 10.1096/fj.201800463.
Comment 10: Fig. 2: Apart from the issue with mtDNA data mentioned above, panel D has no quantification (also panel G). Data in panels E and F are difficult to interpret, since no details are provided for the technique in the first case and for the samples analyzed in the second (the same happens with panel H, which is also not very convincing as there appears to be an artifact in the blot). Please note that these data suggest but not imply (L298) Sirt3 is causing mtHsp70 deacetylation. In fact, based on what is shown in panel G choline produces no change in mtHsp70 acetylation status.
Response 10: Thanks for your valuable comments. Let us answer your concerns one by one:First, we have add the quantification for panel D and G, please see the text.Second, we have provided more detail about determination method for panels E and F, please see the text. As for the panel H, we have provide the original blots, and we have checked and confirmed there are no artifact in our blot.Third, we revised our sentence in L298, please see the text.Lastly, the choline have effect on mtHsp70 acetylation status after our statistical analysis of gray value. For the detail, please see the text.
Comment 11: Fig. 3: fractioning in panel A is not described. In panel E there is a lower Sirt3 signal in the mtHsp70 IP, but this may be due to the decreased levels (HA signal in input and Fig. 2H).
Response 11: First, we have described panel A in notes of Fig.3 based on your comments. Second, in order to eliminate the interference of the decreased signal, we also added the analysis of relative value of gray-scale between Flag-HA and Input-HA in Fig.3 E, and also a quantification for Fig. 2H. For the detail, please see the text. For the detail, please see the text.
Comment 12: Fig. 4: I can make no sense of panel A I am afraid. This shows the impact of FA on overall Lys acetylation, however there is no red signal at all in these images. Assuming it is green, there is indeed increased acetylation with FA, which is completely normalized with choline, but there is no point in doing the merge, is there? Mitochondrial rather than nuclear staining would be expected here. Even though this is the effect on Lys acetylation, Fig. 2G shows no effect of choline on mtHsp70 acetylation, as mentioned above.Why is IP required in panel B?In turn, the siRNA experiment does show the effect on of Sirt3 on mtHsp70 acetylation (please note that the text is wrong in L351), where basically downregulation of Sirt3 by siRNA has a similar effect to FA. However, the levels of Sirt3 are barely altered in the latter. This suggests that the effect of FA is not due to lower Sirt3 expression. Interestingly, panel G shows that ATPase activity is further increased by Sirt3 knockdown (although levels are unchanged). In panel I Sirt3 knockdown is said to reduce ubiquitination, however I do not see that, at least in the presence of PR-619, nor do I see a significant change in acetylation.
Response 12: Thanks for your valuable comments. Let us answer your concerns one by one:First, sorry there is a mistake for signal colour. It did as you mentioned the signal is green and we also revised it. Here, we make merge in order to better describe the location and morphology of cells. And similar study for immunofluorescent analysis of acetylation in fish was also been analysed in our lab (Xu et al., 2020). So here we still continue to use this kind of picture analysis in this study.Second, IP is order to prove the direct relation between Sirt3 and mtHsp70. What’s more, IP here indicated Sirt3 deacetylated mtHSP70 in a dose-dependent manner, which can further convince us Sirt3 did have a regulated role for mtHSP70 deacetylation. Also this similar research method was used in our previous study about yellow catfish (Wei et al., 2021).Lastly, you mentioned the Sirt3 knockdown shown no significant effect on mtHsp70 acetylation under FA condition. Here, we think this should be reasonable and logical. Because UPRmt is just an adaptive mechanism in response to mitochondrial damage. This mechanism aim to restores the mitochondrial normal structure and function at the early mitochondrial damage stage (Zhu et al., 2021). So this mechanism always have sensitive response at initial stage (like just pressure from Sirt3 knockdown/or FA in this present study); however, with aggravated functional damage of mitochondria caused by mtHSP70 acetylation (like double pressure from Sirt3 knockdown and FA), all the mitochondrial related proteins will be less sensitive, which is similar to UPRer, which has also appeared in our previous studies of UPRer in yellow catfish (like double pressure from Sirt3 knockdown and FA (Song et al., 2016). Actually, mtHSP70-ATPase activity on the other hand reflects aggravated UPRmt, but it did shown no significant effect between Sirt3 knockdown + FA group and single Sirt3 knockdown group.Thanks again for your so detail comments.
References:
Xu, Y. C. , Xu, Y. H. , Zhao, T. , Wu, L. X. , & Luo, Z. . Waterborne cu exposure increased lipid deposition and lipogenesis by affecting wnt/β-catenin pathway and the β-catenin acetylation levels of grass carp ctenopharyngodon idella. Environ Pollut. 2020. 114420.
Song YF, Luo Z, Zhang LH, Hogstrand C, Pan YX. Endoplasmic reticulum stress and disturbed calcium homeostasis are involved in copper-induced alteration in hepatic lipid metabolism in yellow catfish Pelteobagrus fulvidraco. Chemosphere. 2016. 144:2443-53. Wei X, Hogstrand C, Chen G, Lv W, Song Y, Xu Y, Luo Z. Zn Induces Lipophagy via the Deacetylation of Beclin1 and Alleviates Cu-Induced Lipotoxicity at Their Environmentally Relevant Concentrations. Environ Sci Technol. 2021 Apr 20;55(8):4943-4953. doi: 10.1021/acs.est.0c08609. Zhu L, Zhou Q, He L, Chen L. Mitochondrial unfolded protein response: An emerging pathway in human diseases. Free Radic Biol Med. 2021. 1;163:125-134. Comment 13: Fig. 5: I am afraid I cannot see much in panel A, this should be much improved. Jc-1 data are similar to previous observations in that Sirt3 knockdown has effects akin to those of FA. As commented above, the methods requires explanation.
Response 13: Thanks for your valuable comments. First, we have revised the panel A in Fig.5. Second, in our present study, the Sirt3 did have similar effect with FA as you mentioned. We found FA induced UPRmt main via decreasing the localization of Sirt3 into MT. So when the Sirt3 knockdown will cause similar effect in this present study. Lastly, based on your comments, we have added more detail about methods in our article. For the detail, please see the text.
Comment 14: Fig. 6: again, the text (L413) is inconsistent with the data. FA enhance mtHsp70 – Acox interaction. Panel C shows that si-Sirt3 results in greatly downregulated Acox1 levels, but if we look at panel E, FA causes a comparable reduction of Sirt3 activity but no change at all in Acox1. This is consistent with FA acting by Sirt3 inhibition (rather than downregulation of expression). On the other hand, the si-Sirt3 data confirm that Sirt3 is required for beta-oxidation rather than prove that Sirt3 mediates the choline effect.
Response 14: Thanks for your valuable comments. We have revised inconformity in L413. Sorry, we didn't understand your meaning for panel E. Because we didn't find the data about Acox1 levels in panel. As you mentioned that these data are not enough to prove Sirt3 mediates the choline effect. We are total agree with this, so we revised this related text in our article. For detail, please see the text.
Comment 15: The Discussion is too long. In particular, the first paragraph is expendable. The remainder would be better if substantially condensed. I must say I see again the authors present the data wrongly. As far as I can tell HFD upregulates mtHsp70 and increases acetylation, not the other way around, via Sirt3 exit to the cytoplasm. Increased mtHsp70 seems to inhibit Acox1. However, the authors fail to show choline affecting either. Perhaps Sirt3 is working via other mechanisms. Because Sirt3 knockdown has such a great effect by itself not that much can be deduced from these experiments in my opinion. I was about to suggest adding a diagram showing the findings of the study and highlighting what remains to be known; then I found such a diagram was actually provided, but only at the very end of supplementary material document, and it is not referred to in the text. This diagram is consistent with my interpretation of the data, with some differences. FA are displayed as uptaken by an apparent channel, this should be modified. I was surprised also to see that there is a second file labelled as ‘non published’ which appears to be identical to the supplementary data file, except for this diagram.Also, HFD upregulates the expression of lipogenic genes, this aspect is pretty much ignored in the rest of the article.The authors are unclear regarding the novelty of mtUPR (I guess this is preferable to UPRmt) in connection with HFD.
Response 15: Thanks for your valuable comments. Let us answer your concerns one by one:First, based on your comments, we have simplified the Discussion section. Please see the text.Second, as for the effect of Sirt3 on choline and the underlying mechanism, we are agree with you. So we revised our Result and Discussion section to weaken this part in our article. For detail, please see the text. At the same time, we added discussion about HFD upregulating the expression of lipogenic genes, and also the the novelty of mtUPR in connection with HFD. For detail, please see the text.Lastly, based on your comments we have revised the graphical conclusions and also moved them from supplementary document to text. For detail, please see the text.
Comment 16: It is advisable to produce the original blots.
Response 16: Thanks for your valuable comments. We have provided the original blots. For detail, please see the text.
Minor comments:
Comment 1: Protein symbols should be in capital letters, gene symbols in italics (but not as written in L155 onwards either).
Response 1: We have check all through article about protein/or gene symbols and revised them based on your comments. For detail, please see the text.
Comment 2: L74: lipotropic? L75: NRC, 2011? L83: duplicated twice?Response 2: Thank you very much for your valuable comments. We have revised the mistake in L74, added the references for NRC in L75. And we also check it is right for L83, which have been described in our previous studies (Chen et al., 2019).
References:
Chen G, Wu K, Zhao T, et al. miR-144 Mediates High Fat–Induced Changes of Cholesterol Metabolism via Direct Regulation of C/EBPα in the Liver and Isolated Hepatocytes of Yellow Catfish. J Nutr, 2019. DOI: 10.1093/jn/nxz282.
Comment 3: L83 on: this explanation is confusing, the authors appear to hint that the extra duplication in fish is relevant for susceptibility to a high fat diet (HFD), but that would make fish a less relevant model for other vertebrates, and HFD do produce steatosis generally. So the point of this part is questionable.
Response 3: Thank you very much for your valuable comments. Here, we just want to emphasize some duplicated genes evolved new functions that in turn resulted in novel regulatory mechanisms in fish. Similarly, in our previous studies about HFD also pointed out the extra duplication (Chen et al., 2019; Ling et al., 2019; Song et al., 2020).
References:
Chen G, Wu K, Zhao T, et al. miR-144 Mediates High Fat–Induced Changes of Cholesterol Metabolism via Direct Regulation of C/EBPα in the Liver and Isolated Hepatocytes of Yellow Catfish. J Nutr, 2019. DOI: 10.1093/jn/nxz282.
Ling S C, Wu K, Zhang D G, et al. Endoplasmic Reticulum Stress–Mediated Autophagy and Apoptosis Alleviate Dietary Fat–Induced Triglyceride Accumulation in the Intestine and in Isolated Intestinal Epithelial Cells of Yellow Catfish. J Nutr, 2019. doi: org/10.1093/jn/nxz135.
Song YF, Hogstrand C, Ling SC, Chen GH, Luo Z. Creb-Pgc1α pathway modulates the interaction between lipid droplets and mitochondria and influences high fat diet-induced changes of lipid metabolism in the liver and isolated hepatocytes of yellow catfish. J Nutr Biochem. 2020. doi: 10.1016/j.jnutbio.2020.108364.
Comment 4: L108: it is best to provide the actual reference.
Response 4: We have added the reference based on your comments. For the detail, please see the text.
Comment 5: L121: growth rate and weight gain are calculated in bulk? Also some more detail about these measurements would be advisable. Why only 9 out of 75 fish for structural analysis? What is the size of a 3 g fish and how it influences analysis for biochemical parameters? What do the authors mean exactly in L140? Also please use names consistently (i.e. scientific and normal name of the species).
Response 5: Thanks for your valuable comments. Let us answer your concerns one by one:
First, the growth rate and weight gain are calculated by weight, and the calculation detail was shown in Supplementary Table 2 (Notes).
Second, the number of fish for structural analysis is based on our previous studies (Chen et al., 2019; Ling et al., 2019; Song et al., 2020; Zhao et al., 2020). This number not only ensure the accuracy of data, but also the most economical for experimental expenses.
Third, the size of a 3 g fish is juvenile fish for yellow catfish, and at this stage, the fish have the best growth performance and also suitable for determinate biochemical parameters. Again this size of fish were used in our previous studies (Chen et al., 2019; Ling et al., 2019; Song et al., 2020; Zhao et al., 2020).
Lastly, in order to make sure the repeatability and accuracy in vitro, we selected three fish for sample and mixed them for cell culture in a pool of cells. And we also check the name of the species based on your comments. For the detail, please see the text.
References:
Chen G, Wu K, Zhao T, et al. miR-144 Mediates High Fat–Induced Changes of Cholesterol Metabolism via Direct Regulation of C/EBPα in the Liver and Isolated Hepatocytes of Yellow Catfish. J Nutr, 2019. DOI: 10.1093/jn/nxz282.
Ling S C, Wu K, Zhang D G, et al. Endoplasmic Reticulum Stress–Mediated Autophagy and Apoptosis Alleviate Dietary Fat–Induced Triglyceride Accumulation in the Intestine and in Isolated Intestinal Epithelial Cells of Yellow Catfish. J Nutr, 2019. doi: org/10.1093/jn/nxz135.
Song YF, Hogstrand C, Ling SC, Chen GH, Luo Z. Creb-Pgc1α pathway modulates the interaction between lipid droplets and mitochondria and influences high fat diet-induced changes of lipid metabolism in the liver and isolated hepatocytes of yellow catfish. J Nutr Biochem. 2020. doi: 10.1016/j.jnutbio.2020.108364.
Zhao T, Wu K, Hogstrand C, Xu YH, Chen GH, Wei CC, Luo Z. Lipophagy mediated carbohydrate-induced changes of lipid metabolism via oxidative stress, endoplasmic reticulum (ER) stress and ChREBP/PPARγ pathways. Cell Mol Life Sci. 2020. 77(10):1987-2003. doi: 10.1007/s00018-019-03263-6.
Comment 6: L127: these fishes as twice as big, why? What is the life span of cultured hepatocytes? What are OA and PA?
Response 6: Here, these fish come were obtained from in vivo experiment after two weeks of acclimation. The two weeks of acclimation is make sure fish have a stable physiological state (Juvenile fish are purchased from wild fisheries), similarly with our studies (Ling et al., 2019; Song et al., 2020).The life span of cultured hepatocytes from yellow catfish was about four days. But the best optimal cell state is 48h, which also similar with our studies (Ling et al., 2019; Song et al., 2020; Wei et al., 2021).The reason for choice of OA and PA is based on our previous studies (Chen et al., 2019; Ling et al., 2019; Song et al., 2020). And also because the OA and PA is most commonly fatty acids for aquaculture.
References:
Chen G, Wu K, Zhao T, et al. miR-144 Mediates High Fat–Induced Changes of Cholesterol Metabolism via Direct Regulation of C/EBPα in the Liver and Isolated Hepatocytes of Yellow Catfish. J Nutr, 2019. DOI: 10.1093/jn/nxz282.
Ling S C, Wu K, Zhang D G, et al. Endoplasmic Reticulum Stress–Mediated Autophagy and Apoptosis Alleviate Dietary Fat–Induced Triglyceride Accumulation in the Intestine and in Isolated Intestinal Epithelial Cells of Yellow Catfish. J Nutr, 2019. doi: org/10.1093/jn/nxz135.
Song YF, Hogstrand C, Ling SC, Chen GH, Luo Z. Creb-Pgc1α pathway modulates the interaction between lipid droplets and mitochondria and influences high fat diet-induced changes of lipid metabolism in the liver and isolated hepatocytes of yellow catfish. J Nutr Biochem. 2020. doi: 10.1016/j.jnutbio.2020.108364.
Wei X, Hogstrand C, Chen G, Lv W, Song Y, Xu Y, Luo Z. Zn Induces Lipophagy via the Deacetylation of Beclin1 and Alleviates Cu-Induced Lipotoxicity at Their Environmentally Relevant Concentrations. Environ Sci Technol. 2021 Apr 20;55(8):4943-4953. doi: 10.1021/acs.est.0c08609.
Zhao T, Wu K, Hogstrand C, Xu YH, Chen GH, Wei CC, Luo Z. Lipophagy mediated carbohydrate-induced changes of lipid metabolism via oxidative stress, endoplasmic reticulum (ER) stress and ChREBP/PPARγ pathways. Cell Mol Life Sci. 2020. 77(10):1987-2003. doi: 10.1007/s00018-019-03263-6.
Comment 7: L157: how stable were those genes?
Response 7: These housekeeping genes are selected and used in large amount of our previous studies involved in yellow catfish (Wei et al., 2018; Chen et al., 2019; Ling et al., 2019; Song et al., 2020; Zhao et al., 2020; Wei et al., 2021). The M data for β-actin and gapdh is 0.27, and these two housekeeping genes were also used in previous studies about HFD in yellow catfish (Song et al., 2020). All these can proved these housekeeping genes are very stable for yellow catfish.
References:
Chen G, Wu K, Zhao T, et al. miR-144 Mediates High Fat–Induced Changes of Cholesterol Metabolism via Direct Regulation of C/EBPα in the Liver and Isolated Hepatocytes of Yellow Catfish. J Nutr, 2019. DOI: 10.1093/jn/nxz282.
Ling S C, Wu K, Zhang D G, et al. Endoplasmic Reticulum Stress–Mediated Autophagy and Apoptosis Alleviate Dietary Fat–Induced Triglyceride Accumulation in the Intestine and in Isolated Intestinal Epithelial Cells of Yellow Catfish. J Nutr, 2019. doi: org/10.1093/jn/nxz135.
Song YF, Hogstrand C, Ling SC, Chen GH, Luo Z. Creb-Pgc1α pathway modulates the interaction between lipid droplets and mitochondria and influences high fat diet-induced changes of lipid metabolism in the liver and isolated hepatocytes of yellow catfish. J Nutr Biochem. 2020. doi: 10.1016/j.jnutbio.2020.108364.
Wei CC, Luo Z, Hogstrand C, Xu YH, Wu LX, Chen GH, Pan YX, Song YF. Zinc reduces hepatic lipid deposition and activates lipophagy via Zn2+/MTF-1/PPARα and Ca2+/CaMKKβ/AMPK pathways. FASEB J. 2018. 18:fj201800463. doi: 10.1096/fj.201800463.
Wei X, Hogstrand C, Chen G, Lv W, Song Y, Xu Y, Luo Z. Zn Induces Lipophagy via the Deacetylation of Beclin1 and Alleviates Cu-Induced Lipotoxicity at Their Environmentally Relevant Concentrations. Environ Sci Technol. 2021 Apr 20;55(8):4943-4953. doi: 10.1021/acs.est.0c08609.
Zhao T, Wu K, Hogstrand C, Xu YH, Chen GH, Wei CC, Luo Z. Lipophagy mediated carbohydrate-induced changes of lipid metabolism via oxidative stress, endoplasmic reticulum (ER) stress and ChREBP/PPARγ pathways. Cell Mol Life Sci. 2020. 77(10):1987-2003. doi: 10.1007/s00018-019-03263-6.
Comment 8: L169: if a kit for measuring deacetylase activity is used, this is what ought to be reported, not Sirt3 activity. L171: please describe assay briefly. L172: what do the authors mean by ‘approximate single turnover measurements’?
Response 8: We have change SIRT3 deacetylase activity instead of Sirt3 activity in L169. For L171, we describe ACOX1 enzyme activity assay based on your comment. We added more detail about HSP70-ATPase activity measurements in according to published methods in L172 (Fewell et al., 2004; Sullivan et al., 2000).Thanks again for so detail comments! For the detail, please see the text.
References:
Fewell SW, Smith CM, Lyon MA, Dumitrescu TP, Wipf P, Day BW, Brodsky JL. Small molecule modulators of endogenous and co-chaperone-stimulated hsp70 ATPase activity. J Biol Chem. 2004; 279:51131-51140.Sullivan CS, Tremblay JD, Fewell SW, Lewis JA, Brodsky JL, Pipas JM. Species-specific elements in the large T-antigen J domain are required for cellular transformation and DNA replication by simian virus 40. Mol Cell Biol. 2000 Aug;20(15):5749-57. doi: 10.1128/MCB.20.15.5749-5757.2000.
Comment 9: L195: provide details of equipment used.
Response 9: We have provided details of equipment based on your comment.
Comment 10: What was the efficiency of siRNA transfection?
Response 10: Sorry, here, we didn't determine the detail efficiency of siRNA transfection. But we have shown the inhibition efficiency of siRNA on Supplementary Fig. 2. B-D.
Comment 11: L286: it is decreased; L287: but?
Response 11: We have revised them based on your comment. For detail, please see text.
Comment 12: Fig. 4C: what do the authors make of the 2KR signal being so similar to that of the single mutants?
Response 12: In Fig. 4C, the 2KR signal was lower than single mutants after statistical analysis of gray value, but no so notable. The same situation was also appeared in our previous study about yellow catfish (Wei et al., 2021).
References:
Wei X, Hogstrand C, Chen G, Lv W, Song Y, Xu Y, Luo Z. Zn Induces Lipophagy via the Deacetylation of Beclin1 and Alleviates Cu-Induced Lipotoxicity at Their Environmentally Relevant Concentrations. Environ Sci Technol. 2021 Apr 20;55(8):4943-4953. doi: 10.1021/acs.est.0c08609. Comment 13: Revise labeling in Y axis, panel 6F.
Response 13: We have revised the labeling for Y axis in panel 6F based on your comment. For detail, please see text.
Comment 14: Please avoid the expression ‘upregulation of acetylation’.
Response 14: We check all this expression in article and revised them based on your comment. For detail, please see text.

Round 2
Reviewer 1 Report
My comments have been addressed.
Author Response
My comments have been addressed.
Response: Thanks your valuable and kind comments. All your comments will helpful for our future study.
Reviewer 2 Report
I have concluded my second revision of this manuscript. It is unfortunate that the authors did not highlight the changes in the manuscript (and that the journal did not require them to do so). This has complicated this revision to a substantial extent, especially since the changes abound. I appreciate the authors’ efforts to meet my suggestions, however I still have a number of comments, as follows.
- The effort made by the authors is patent, yet the present form is still plagued with English errors, and beyond that, it is difficult to read overall. It is nonetheless clearer in terms of readability. But I recommend professional editing here.
- The fact that choline has the reported effects makes one wonder what happens in basal conditions. According to the authors, this question has been answered previously in their experimental model. This is acceptable, but I wish the main issues would have been brought forward in the discussion.
- The figures are much better now, but issues remain. Labeling of Fig. 1B is incorrect (what the authors show here is vacuoles). The authors missed my point entirely regarding the labeling of panel A, by the way, what I meant was that, as they label he and va separately, it looks as though vacuoles exist independently of hepatocytes.
- Page 9, paragraph 2: the downregulation of SIRT3 does not imply anything in the figure, it does suggest a mechanistic involvement, but this is unproven at this point. Also in the first paragraph, RNA is increased by HFD, not decreased. The blots shown in panels H and J are problematic; in the first case the signal is almost identical in lanes 2 and 3 (as I stated in my first review), implying that it is not the most representative blot of the newly added quantification. In the second, I see a defect in the blot (possible transfer problem centered on the second lane but extending left and right). The authors claim they have checked for artifacts – how? In a blot, if there is a band with uneven intensity side to side this is normally an artifact, particularly if this extends to the neighboring lane.
- Page 11, line 1: again, the data do not support the claim, what panel A in Fig. 3 establishes is that SIRT3 is expressed in the mitochondria (all these are proteins, by the way, so they should be written in capital letters). Fractioning process remains undescribed. It would be nice to know what samples were analyzed in HepG2 cells (it looks like whole cell homogenates).
- The authors base their use of antibodies in previous publications, which they cite in the response, but apparently not in the manuscript.
- Fig. 4: I still see no point in the merge, but that is of no consequence. Despite the authors’ claim, wouldn’t a standard WB do the same (panel B)? That is, if the antibody used for staining recognizes acetylated mtHsp70 as shown in Fig. 2H; if it binds acetylated lysine on the other hand, it makes total sense. We cannot know because the paper does not tell one way or the other. That is why I asked about this. At any rate, in this experiment SIRT3 was overexpressed to different levels and mtHsp70 acetylation, but we get no data confirming the former (i.e. HA signal), which I asked for.
The authors state that FA decreases SIRT3 levels, leading to increased mtHsp70 acetylation. The problem is that, while the blot does suggest some downregulation, it looks minimal, if at all. And I already am unconvinced by the previous blot in Fig. 2J. Certainly nothing like the reduction caused by siRNA. It could be that the effect on acetylation saturates with little decreases of SIRT3, but overall I find these data doubtful for the interpretation given. The fact that ATPase activity is further increased by Sirt3 knockdown (although levels are unchanged) independently of mtHsp70 acetylation is indicative that some of SIRT3 actions differ mechanistically.
The authors include this sentence in their answer. ‘Lastly, you mentioned the Sirt3 knockdown shown no significant effect on mtHsp70 acetylation under FA condition. Here, we think this should be reasonable and logical’. To be clear, I never said that, mtHsp70 acetylation is indeed increased by FA, and what I said is that I do not see (currently) any effect at this level of choline in Fig. 2H. So I am puzzled by the comments that follow (which I understand to be contrary to their own original interpretation).
The authors did not respond to my comment on ubiquitination (panel I). These data are problematic. On the one hand, SIRT3 knockdown does show less ubiquitination and augmented acetylation. The interpretation would be that acetylation inhibits ubiquitination (not that there is a competitive interaction). But then, on the other hand, when deubiquitination is inhibited, ubiquitin signal is increased as expected, but equally in both groups, which is surprising although possible. The question here is that the acetylation signal is not that clear, the authors see downregulation, I have my doubts because it looks quite similar to the band in lane 2. This has a good solution: do more experiments and provide the statistics. Please note that if the authors’ interpretation is true, ubiquitination modulates acetylation. Then why the band is unchanged in lane 3 vs. 1?
- Fig. 2, panel A, left: are the authors sure the arrow points at a mitochondrion?
- Fig. 5A remains the same this is completely useless in present form in my opinion.
- Fig. 6: My interpretation of these data is that FA enhance mtHsp70 – Acox interaction and that this correlates with reduced SIRT3 function and mtHSP70 acetylation, plus downregulation of catabolic genes. Because si-Sirt3 kills everything it results that SIRT3 is required for catabolism. If we look at the WB in panel C we see that SIRT3 expression is unchanged. Thus the mechanism is related to inhibition of SIRT3 rather than reduced expression. This is inconsistent with Fig. 3B by the way. So it is next to impossible to tell what is really happening in terms of expression. At any rate, reduced expression also works because SIRT3 is critical. It blocks choline effects on beta-oxidation but this shows nothing, because there is no beta-oxidation in these conditions. Choline however increases SIRT3 function, so this may be the mechanism. The authors claim that this is related to increased localization in mitochondria, but we do not know what samples were used for the activity assay (if it is whole cells it tells us nothing in terms of localization), and Fig. 3B is difficult to interpret, in that there is decreased signal, yes, but localization does not apparently change, and as mentioned this is inconsistent with WB data.
- It is advisable to produce the original blots. I do not see them.
- 2.2.8: spermatozoa?
- Page 12, line 10: concentration, not dose.
- The authors still fail to define OA and PA.
Author Response
I have concluded my second revision of this manuscript. It is unfortunate that the authors did not highlight the changes in the manuscript (and that the journal did not require them to do so). This has complicated this revision to a substantial extent, especially since the changes abound. I appreciate the authors’ efforts to meet my suggestions, however I still have a number of comments, as follows.
Response: Here, sorry there are no highlight for the changes in the manuscript. But when I submitted manuscript the revision mode as word was provided. So this time I will highlight all the change (red marker) for your comments.
Comment 1: The effort made by the authors is patent, yet the present form is still plagued with English errors, and beyond that, it is difficult to read overall. It is nonetheless clearer in terms of readability. But I recommend professional editing here.
Response 1: Although last time we have revised our English errors, the quality of writing still needs to be improved. So based on your comments, we do our best to check these errors and revise modification again this time, meanwhile, we compressed the whole article. All the change have been marked by red colour. We hope this version can make you read more smoothly. Thank you for your valuable comments on the writing for this article.
Comment 2: The fact that choline has the reported effects makes one wonder what happens in basal conditions. According to the authors, this question has been answered previously in their experimental model. This is acceptable, but I wish the main issues would have been brought forward in the discussion.
Response 2: We have discussed the issues in the discussion section based on your comments. For the detail, please see the text.
Comment 3: The figures are much better now, but issues remain. Labeling of Fig. 1B is incorrect (what the authors show here is vacuoles). The authors missed my point entirely regarding the labeling of panel A, by the way, what I meant was that, as they label he and va separately, it looks as though vacuoles exist independently of hepatocytes.
Response 3: Thanks for your valuable comments. First, we have revised the labeling of Fig. 1B, please see the text. Second, in this present study, the vacuoles exist dependently of hepatocytes. For yellow catfish, the vacuolation in hepatocytes was very serious after HFD, so that it seems have filled the whole cell and exist independently of hepatocytes. This similar result was also have been shown in other study in our lab (Chen et al., 2019).
References:
Chen G, Wu K, Zhao T, et al. miR-144 Mediates High Fat–Induced Changes of Cholesterol Metabolism via Direct Regulation of C/EBPα in the Liver and Isolated Hepatocytes of Yellow Catfish. J Nutr, 2019. DOI: 10.1093/jn/nxz282.
Comment 4: Page 9, paragraph 2: the downregulation of SIRT3 does not imply anything in the figure, it does suggest a mechanistic involvement, but this is unproven at this point. Also in the first paragraph, RNA is increased by HFD, not decreased. The blots shown in panels H and J are problematic; in the first case the signal is almost identical in lanes 2 and 3 (as I stated in my first review), implying that it is not the most representative blot of the newly added quantification. In the second, I see a defect in the blot (possible transfer problem centered on the second lane but extending left and right). The authors claim they have checked for artifacts – how? In a blot, if there is a band with uneven intensity side to side this is normally an artifact, particularly if this extends to the neighboring lane.
Response 4: Thanks for your valuable comments. Let us answer your concerns one by one:First, we have deleted content in Page 9, paragraph 2. Meanwhile, we revised our text in the first paragraph. For the detail, please see the text.Second, we check our all bolt by the original blots. But, here, for panel H and J in Fig. 2, the bolt did seems no significant change and there are a defect in J as you mentioned. And we also agree with you, so we provided other representative blot for H and J in Fig. 2. For the detail, please see the text.Thanks again for your so detail comments.
Comment 5: Page 11, line 1: again, the data do not support the claim, what panel A in Fig. 3 establishes is that SIRT3 is expressed in the mitochondria (all these are proteins, by the way, so they should be written in capital letters). Fractioning process remains undescribed. It would be nice to know what samples were analyzed in HepG2 cells (it looks like whole cell homogenates).
Response 5: Thanks for your valuable comments. We have deleted content in Page 11, line 1. At the same time, we have revised all these are proteins in capital letters even in Figures based on your comments. Fractioning process was also added in Result section, and also the samples analyse for HepG2 cells. For the detail, please see the text.
Comment 6: The authors base their use of antibodies in previous publications, which they cite in the response, but apparently not in the manuscript.
Response 6: We have added these information in the manuscript based on your comments.
Comment 7: Fig. 4: I still see no point in the merge, but that is of no consequence. Despite the authors’ claim, wouldn’t a standard WB do the same (panel B)? That is, if the antibody used for staining recognizes acetylated mtHsp70 as shown in Fig. 2H; if it binds acetylated lysine on the other hand, it makes total sense. We cannot know because the paper does not tell one way or the other. That is why I asked about this. At any rate, in this experiment SIRT3 was overexpressed to different levels and mtHsp70 acetylation, but we get no data confirming the former (i.e. HA signal), which I asked for.The authors state that FA decreases SIRT3 levels, leading to increased mtHsp70 acetylation. The problem is that, while the blot does suggest some downregulation, it looks minimal, if at all. And I already am unconvinced by the previous blot in Fig. 2J. Certainly nothing like the reduction caused by siRNA. It could be that the effect on acetylation saturates with little decreases of SIRT3, but overall I find these data doubtful for the interpretation given. The fact that ATPase activity is further increased by Sirt3 knockdown (although levels are unchanged) independently of mtHsp70 acetylation is indicative that some of SIRT3 actions differ mechanistically.The authors include this sentence in their answer. ‘Lastly, you mentioned the Sirt3 knockdown shown no significant effect on mtHsp70 acetylation under FA condition. Here, we think this should be reasonable and logical’. To be clear, I never said that, mtHsp70 acetylation is indeed increased by FA, and what I said is that I do not see (currently) any effect at this level of choline in Fig. 2H. So I am puzzled by the comments that follow (which I understand to be contrary to their own original interpretation).The authors did not respond to my comment on ubiquitination (panel I). These data are problematic. On the one hand, SIRT3 knockdown does show less ubiquitination and augmented acetylation. The interpretation would be that acetylation inhibits ubiquitination (not that there is a competitive interaction). But then, on the other hand, when deubiquitination is inhibited, ubiquitin signal is increased as expected, but equally in both groups, which is surprising although possible. The question here is that the acetylation signal is not that clear, the authors see downregulation, i have my doubts because it looks quite similar to the band in lane 2. This has a good solution: do more experiments and provide the statistics. Please note that if the authors’ interpretation is true, ubiquitination modulates acetylation. Then why the band is unchanged in lane 3 vs. 1?
Response 7: Thanks for your valuable comments. Let us answer your concerns one by one:First, we have deleted the merge in Fig. 4 A based on your comments. As for the antibody, in this present study, the antibody binding acetylated lysine. In in Fig. 2H, by IP pulling down mtHSP70, then we used the antibody to recognize acetylated lysine. We also describe this information in our text. On the other hand, as you mentioned SIRT3 overexpressed experiment, we have added HA signal in Fig. 4 B to confirm SIRT3 overexpression. Second, we have provide new representative blot in Fig. 2J, and hope can explain the increased mtHsp70 acetylation caused by HFD. As for the effect of SIRT3 knockdown on mtHsp70 acetylation, we agree with your point, so in this present we revised text and didn't claim the direct regulatory role of Sirt3 on mtHsp70 acetylation. Also we have deleted the data about ATPase activity after Sirt3 knockdown, which cann’t explain the relation between ATPase activity and Sirt3.Third, sorry here we misunderstand your meanings. But for Fig. 2H, we have provided new representative blot, which shown the effect of choline on mtHsp70 acetylation. So in our present study, we think HFD can increase mtHsp70 acetylation but choline can induce relieve effect.Lastly, in term of ubiquitination, we sorry miss response your comments last time. We are agree with your view. The data in Fig. 4 I seems no enough to indicate the ubiquitination modulates acetylation. Considering these data was not core important data, so we deleted the merge in Fig. 4 I and change all our statement about this based on your comments. At the same time we will try repeat this part of the experiment based on your comments but here we cann’t provide our new validation data due to the deadline for resubmit (7 days). For the detail, please see the text.Here we hope your main concerns can be answered, and also if you still think there are some defect for this part, please do not hesitate to contact us. Thanks again for your comments!
Comment 8: Fig. 2, panel A, left: are the authors sure the arrow points at a mitochondrion?
Response 8: The arrow points at mitochondrion after our check. These mitochondrion doesn't look like traditional mitochondrial structure. Here these mitochondrion should be in a highly fused state, so this was not a mitochondrion and should be a group of mitochondrion.
Comment 9: Fig. 5A remains the same this is completely useless in present form in my opinion.
Response 9: We have revised the Fig. 5A based on your comments. This time we partially enlarged representative mitochondrion in TEM. For the detail, please see the text.
Comment 10: Fig. 6: My interpretation of these data is that FA enhance mtHsp70 – Acox interaction and that this correlates with reduced SIRT3 function and mtHSP70 acetylation, plus downregulation of catabolic genes. Because si-Sirt3 kills everything it results that SIRT3 is required for catabolism. If we look at the WB in panel C we see that SIRT3 expression is unchanged. Thus the mechanism is related to inhibition of SIRT3 rather than reduced expression. This is inconsistent with Fig. 3B by the way. So it is next to impossible to tell what is really happening in terms of expression. At any rate, reduced expression also works because SIRT3 is critical. It blocks choline effects on beta-oxidation but this shows nothing, because there is no beta-oxidation in these conditions. Choline however increases SIRT3 function, so this may be the mechanism. The authors claim that this is related to increased localization in mitochondria, but we do not know what samples were used for the activity assay (if it is whole cells it tells us nothing in terms of localization), and Fig. 3B is difficult to interpret, in that there is decreased signal, yes, but localization does not apparently change, and as mentioned this is inconsistent with WB data.
Response 10: Thanks for your valuable comments. Let us answer your concerns one by one:First, in panel C of Fig. 6, WB shown ACOX1 expression, not SIRT3. The SIRT3 expression was decreased by si-Sirt3 at both mRNA and protein levels, and these result was shown in Supplementary Figure. 1. However, here, we are total agree with your point about the si-Sirt3 kills everything and then result in the downregulation of catabolic genes. So in our Discussion section we discussed this potential mechanism based on your comments.Second, the samples were after the isolation and purification of mitochondria were samples for activity assay in this present. And we have added this information in our text. On the other hand, in this present study co-localization analysis of SIRT3 and mitochondria was marked by both SIRT3 and mitochondria. In theory, these result could reflect the level of SIRT3 in mitochondria. Similarly this co-localization analysis was also been tested in our previous study (Song et al., 2020). Importantly, we agree with your view about potential mechanism of choline increasing SIRT3 function, not just by expression. Also we added this content in our Discussion section. For the detail, please see the text. Thanks for your detail comments! References:
Song YF, Hogstrand C, Ling SC, Chen GH, Luo Z. Creb-Pgc1α pathway modulates the interaction between lipid droplets and mitochondria and influences high fat diet-induced changes of lipid metabolism in the liver and isolated hepatocytes of yellow catfish. J Nutr Biochem. 2020. doi: 10.1016/j.jnutbio.2020.108364.
Comment 11: It is advisable to produce the original blots. I do not see them.
Response 11: Last time we have provided the original bolts in one PDF document and submitted them by Figures in window options of DMPI system. But I didn't know why you cann't see them. Anyway I will submit them again this time.
Comment 12: 2.2.8: spermatozoa?
Response 12: We have revised them to hepatocytes and thanks for your comments.
Comment 13: Page 12, line 10: concentration, not dose.
Response 13: We have revised it based on your comments.
Comment 14: The authors still fail to define OA and PA.
Response 14: We have define OA for oleic acid and PA for palmitic acid. For the detail, please see the text.
